# Aerosol formation and growth rates from chamber experiments using Kalman smoothing

Matthew Ozon[1], Dominik Stolzenburg[2], Lubna Dada[2,3,4], Aku Seppänen[1] and Kari E.J. Lehtinen[1,5]

[1] Department of Applied Physics, University of Eastern Finland, Kuopio, Finland

[2] Institute for Atmospheric and Earth System Research/ Physics, University of Helsinki, 00014 Helsinki, Finland

[3] EPFL, School of Architecture, Civil and Environmental Engineering, Sion, 1951, Switzerland

[4] Paul Scherrer Institute, Laboratory of Atmospheric Chemistry, 5232 Villigen PSI, Switzerland

[5] Atmospheric Research Centre of Eastern Finland, Finnish Meteorological Institute, Kuopio, Finland

*Correspondence to*: Kari E.J. Lehtinen (kari.lehtinen@uef.fi)

Keywords. Particle number size distribution, nucleation, growth, chamber experiments, Bayesian state estimation, Kalman smoother, DMA train

**Abstract.**

Bayesian state estimation in the form of Kalman smoothing was applied to Differential Mobility Analyser Train (DMA-train) measurements of aerosol size distribution dynamics. Four experiments were analysed in order to estimate the aerosol size distribution, formation rate and size-dependent growth rate, as functions of time. The first analysed case was a synthetic one, generated by a detailed aerosol dynamics model, and the other three chamber experiments performed at the CERN CLOUD facility. The estimated formation and growth rates were compared with other methods used earlier for the CLOUD data and with the true values for the computer-generated synthetic experiment. The agreement in the growth rates was very good for all studied cases: estimations with an earlier method fell within the uncertainty limits of the Kalman smoother results. The formation rates matched also well, within roughly a factor of 2.5 in all cases, which can be considered very good considering the fact that they were estimated from data given by two different instruments, the other being the Particle Size magnifier (PSM), which is known to have large uncertainties close to its detection limit. The presented Fixed Interval Kalman Smoother (FIKS) method has clear advantages compared with earlier methods that have been applied to this kind of data. First, FIKS can reconstruct the size distribution between possible size gaps in the measurement in such a way that it is consistent with aerosol size distribution dynamics theory, and second, the method gives rise to direct and reliable estimation of size distribution and process rate uncertainties if the uncertainties in the kernel functions and numerical models are known.

## 1 Introduction

Atmospheric new particle formation and growth are important phenomena when considering global aerosol concentrations. Aerosol number concentration together with their size distribution and chemical composition determines how aerosols affect visibility, health and climate (Albrecht, 1989; Appel et al., 1985; Daellenbach et al., 2020; Pope and Dockery, 2006; Twomey, 1974). These are determined by atmospheric dynamics and aerosol dynamics such as new particle formation and growth as well as removal rates. Nieminen et al. (2018) reviewed the existing literature on the formation and growth rates ranging from polar sites, with very small aerosol concentrations to polluted urban sites with

extremely high concentrations. The rates have been typically estimated using the methodology reviewed in Kulmala et al. (2012) and adjusted for chamber experiments by Dada et al. (2020). Both are based on rather simple regression or balance equation approaches, and permitting no proper estimation of the uncertainties. At the same time, however, instrument development, especially advances in particle detection efficiency and mass spectrometry, has developed rapidly (Kangasluoma et al., 2020). Potentially superior advanced data analysis methods have not been used, and, it is likely that there are significant inaccuracies in the estimated particle formation and growth rates estimated previously (Kürten et al., 2018).

There have been some attempts to estimate aerosol formation and growth rates with different inverse methods (Henze et al., 2004; Kuang et al., 2012; Lehtinen et al., 2004; Sandu et al., 2005; Verheggen and Mozurkewich, 2006; Viskari et al., 2012). We are, however, not aware of any of the above-mentioned methodology being used widely. The most promising ones in our view, that include also estimations of uncertainties, have been the ones by Kupiainen-Määttä (2016) and Shcherbacheva et al. (2020), who used Markov Chain Monte Carlo methodology to estimate evaporation rates as well as their uncertainties from synthetic cluster dynamics data. In addition, the INSIDE-method by Pichelstorfer et al. (2018), which is based on numerical solution of the aerosol general dynamic equation and matching the solution optimally to integrated measured concentrations of selected size intervals, has been successfully applied to determining growth rates at the CLOUD (Cosmics Leaving OUtdoor Droplets) experiments at CERN (European Organization for Nuclear Research) (Stolzenburg et al., 2020). Furthermore, the results in very recent manuscript by McGuffin et al. (2020), in which nucleation, growth and emission rates of are estimated using techniques from the field of nonlinear process control, seem promising.

In a recent paper, Ozon et al. (2020) presented BAYROSOL, a Julia software package that combines a finite difference solution to the general dynamic equation for aerosols (GDE; Seinfeld and Pandis, 2016) to Bayesian state estimation in order to estimate unknown size dependent process rates (nucleation, condensation, losses) from known time evolution of the aerosol size distribution. Bayesian state estimation is a general framework for estimating time-dependent variables (state variables) based on (direct or indirect) noisy observations that are collected sequentially during the temporal evolution of the state variables (Gelb, 1979). The state estimation is based on the so-called state-space representation, which consists of the state evolution model and observation model. In this work, the state variables consist of the particle size distribution – the temporal evolution of which is modelled with GDE – and the nucleation, growth and deposition rates which are parameters of the GDE. The observation model is the mapping from the size distribution to DMA-train measurements. In Bayesian formulation, both the state variable and the observations are modelled as stochastic processes; their randomness reflects their uncertainty, which decreases, when measurement data is accounted for in the state estimation – formally speaking, this is done by conditioning the state variables with respect to measurement data (realized observations) sequentially. The result of Bayesian state estimation is the posterior probability density which reflects the uncertainty of the state variables after accounting for the measurement data.

A large variety of state estimation schemes exits, and the choice between them depends on 1) the type of the state-space model (linearity, gaussianity, etc); 2) the type of data available when computing an estimate at time t (if data is available up to time $k < t$, the problem is of prediction type, while cases where $k = t$ and $k > t$ are referred to as filtering and smoothing, respectively), and 3) the approximations which are sometimes needed to lower the computational demand of state estimation. In the case of linear and gaussian state-space models, a Bayesian filtering problem can be solved recursively by the well-known Kalman filter algorithm. In non-linear and non-gaussian cases, the rigorous choice is to use so-called particle filters and smoothers (Särkkä, 2013). However, because these MCMC-based estimators are highly time

consuming in large-dimensional cases, approximative methods are often used – such as the extended Kalman filter and smoother adopted in this paper. These recursive algorithms use sequential linearization to approximate the non-linear models and non-gaussian probability distributions.

In previous work Ozon et al., (2020) showed that the Fixed Interval Kalman Smoother (FIKS; Kaipio and Somersalo, 2005) performed very well in estimating the process rates of the GDE in two distinct cases. First in a case in which continuous nucleation, growth and losses lead to a nearly steady state size distribution and second also in a case in which there is a growing nucleation mode after a nucleation burst. In the method, the unknowns (such as the discretized particle size distribution) are modeled as random variables, and their prior probability distributions are incorporated in the solution of the inverse problem. One important key feature of the Kalman smoother method is that it estimates also the error covariance matrices of the process rates that is their uncertainties, if the uncertainties of the measurement device are known.

In this manuscript, we show results of applying BAYROSOL, for the first time to real experimental data. We use experiments performed with the differential mobility analyser-train (DMA-train; Stolzenburg et al., 2017) measuring new particle formation and growth at the CERN CLOUD chamber (Duplissy et al., 2016; Kirkby et al., 2011). In addition to testing the method with synthetic DMA-train data (in which the 'correct' results are known), we estimate formation and growth rates from three different formation and growth experiments: sulphuric-ammonia (Stolzenburg et al., 2020), alpha-pinene ozonolysis (Heinritzi et al., 2020) and iodic acid (He et al., 2021; Sipilä et al., 2016). We compare the formation rates with results obtained by using the methodology by Dada et al. (2020) based on Particle Size Magnifier (PSM) measurements and the growth rates with the results obtained by the INSIDE method (Pichelstorfer et al., 2018). We chose the DMA-train measurements for three main reasons: first, the high time resolution makes it an ideal instrument for nucleation studies due to a more accurate estimate of dN/dt. Second, the collection efficiencies of the channels have been carefully characterized (Stolzenburg et al., 2017; Wlasits et al., 2020) and yield higher sensitivities to low particle number concentrations (Kangasluoma et al., 2020), which are often faced in atmospherically relevant nucleation studies, and third, the DMA train is, at the same time, interesting and challenging instrument for detailed data analysis because of the gaps in the measured size spectrum.

## 2    Bayesian framework for parameter estimation

### 2.1    Aerosol  measurement and evolution models

Typical quantities of interest in chamber experiments studying new particle formation are the particle number size-distribution, the formation rate at the critical cluster size and the growth rate of the freshly formed particles. The available measurements to infer these quantities usually size classify the aerosol and measure the size classified number concentrations. While the retrieval of the particle number size distribution from such measurements is a classical inverse problem (Chambolle and Pock, 2011; Fiebig et al., 2005; Kandlikar and Ramachandran, 1999; Wolfenbarger and Seinfeld, 1990), the estimation of the process parameters (formation and growth rate) is often done by analysis of the time evolution of the retrieved particle size distribution (Dada et al., 2020). Here, we focus on the formulation of the problem within a statistical Bayesian framework, where the state parameters described by a measurement model and an evolution model are treated as multivariate random processes and are estimated from measurements using a FIKS (Ozon et al., 2020).

A measurement can be described by a vector $y^k$ representing $m$ indirect observations (channels of the instrument) of the
particle size distribution $n(d_p, t_k)$. The observations are linked to the size distribution by the so-called instrument transfer
(or kernel) functions $\mathcal{H}$ such that:
$$y^k = \int \mathcal{H}(d_p) n(d_p, t_k) dd_p \qquad (1)$$
The transfer functions $\mathcal{H}$ can be inferred from calibration experiments and instrument design considerations. Direct
inversion of Eq. (1) for every time instant $t_k$ is typically an underdetermined and ill-posed inverse problem and requires
some additional assumptions in order to avoid oscillatory and unstable solutions (Kandlikar and Ramachandran, 1999).
At the same time, the time evolution of the particle size-distribution $n(d_p, t)$ can be described by the aerosol general
dynamic equation (GDE):
$$\frac{\partial n}{\partial t}(d_p, t) = -\frac{\partial g(d_p,t)n(d_p,t)}{\partial d_p} - \lambda(d_p, t)n(d_p, t) - \text{CoagSink}(\beta, d_p, t) + \text{CoagSrc}(\beta, d_p, t) \qquad (2)$$
Here, $g(d_p, t)$ is the condensation growth/evaporation rate, $\lambda(d_p, t)$ is the particle loss by deposition or dilution and
CoagSink and CoagSrc are the sink and source rates due to particle coagulation within the size distribution with the
coagulation coefficients $\beta$. An exact expression of all terms can be found in e.g. Ozon et al. (2020) and Seinfeld and
Pandis (2016). The boundary conditions of Eq. (2) are given by the apparent formation rate $J_{d_{min}}(t) =$
$g(d_{min}, t)n(d_{min}, t)$ of newly formed particles at the minimum detectable size $d_{min}$ and a zero numerical flux condition
$g(d_\infty, t)n(d_\infty, t) = 0$ at very large sizes. Altogether, the process parameters $g(d_p, t), J(t), \lambda(d_p, t)$ and $\beta(d_i, d_j)$ as well
as the initial and boundary conditions determine completely the evolution of the size distribution, but especially $g(d_p, t)$
and $J(t)$ are usually not known. The coagulation coefficients $\beta(d_i, d_j)$ can often be obtained from theory (and coagulation
above $d_{min}$ can be even neglected in many applications with low particle concentrations) and the loss parameters $\lambda(d_p, t)$
are well quantified for controlled aerosol chamber experiments.
A single measurement of the size distribution $y^k$ does not depend explicitly on the process parameters, but as $g(d_p, t)$
and $J(t)$ determine the temporal evolution of $n(d_p, t_k)$ the estimation of the process parameters is feasible from a
sequence of $l$ measurements $y^k$ at several time instances.
**2.2   State estimation with Kalman smoothing**
Following Ozon et al. (2020) we formulate the problem as a Bayesian state estimation problem. After discretization of
the problem in size space, i.e. particle diameter ($i = 1, .., q$) and time ($k = 1, ... l$), we can define the state variable $X^k =$
$\begin{bmatrix} N^k & g^k & \lambda^k & J^k \end{bmatrix}^T$ for each time step $k$ with the particle concentrations $N_i^k$ per size discretization bin $i$, the
condensation and loss terms $g_i^k$ and $\lambda_i^k$, respectively, for each size discretization bin $i$ and the nucleation rate $J^k$. Here,
we have denoted $N^k = [N_1^k, ..., N_q^k], N^k = [g_1^k, ..., g_q^k]$ and $\lambda^k = [\lambda_1^k, ..., \lambda_q^k]$. The problem can then be formulated as:
$$X^{k+1} = F(X^k) + w^k \qquad (3)$$
$$y^k = HX^k + v^k \qquad (4)$$
Eq. (3) represents the discretized non-linear evolution model, which is based on the general dynamic equation for $N^k$, on
second order processes for $g^k$ and $J^k$, and a random walk evolution for $\lambda^k$ (see Section 2.3, Eq (5)-(7)). Eq. (4) represents
the discretized linear observation model. The terms $w^k$ and $v^k$ are the error terms, which are approximated as normally
distributed $\mathcal{N}(0, \Gamma_w^k)$ and $\mathcal{N}(0, \Gamma_e^k)$ with the covariance matrices $\Gamma_w^k, \Gamma_v^k$, which not only include stochastic noise, but also
errors due discretization, model and parameter uncertainties.

We note that the above description of the state-space model (3)-(4) is slightly simplified for the sake of notational convenience. Namely, two additional features – both described in detail by Ozon et al. (2020) – are included in the model: First, we assume that the process rates are positive quantities and incorporate this positivity constraint into the evolution model by reparametrizing these quantities in the model. For example, for the nucleation rate $J^k$, we write $J^k = \frac{1}{\alpha}\ln(1 + e^{\alpha\xi_J^k})$, where $\xi_J^k$ is an unconstrained random variable and $\alpha$ is a scaling constant. Respective parametrizations are written for $g_i^k$ and $\lambda_i^k$. Secondly, as noted above, second order models are written for rates $J^k$ and $g^k$. More specifically, we consider the respective state parameters $\xi_J^k$, $\xi_g^k$ as second order Markov processes; for example $\xi_J^k = \psi_1\xi_J^{k-1} + \psi_2\xi_J^{k-2} + \eta$, where $\psi_1$ and $\psi_2$ are model parameters and $\eta$ is Gaussian state noise. The second order models are written, because they promote temporal smoothness of the processes. When the positivity constraint and the second order models are included in the model, the state variable $X^k$ in the state-space model (3)-(4) is rewritten in the form $X^k = \begin{bmatrix} N^k & \xi_g^k & \xi_g^{k-1} & \lambda^k & \xi_J^k & \xi_J^{k-1} \end{bmatrix}^T$, and at each time step the above logarithmic functions are used for mapping the unconstrained variables $\xi_J^k$, $\xi_g^k$ and $\xi_\lambda^k$ to respective quantities $J^k, g_i^k$ and $\lambda_i^k$. For the details on the above modifications as well as discretization of the GDE model, we refer to Ozon et al. (2020).

The GDE, i.e. the non-linear evolution model for $N^k$ (Eq. (2)), is similar to an advection equation. Therefore, its numerical discretization schemes are often unstable and must be treated carefully to avoid oscillation and divergence or to minimize numerical diffusion (Shen et al., 2007; Smolarkiewicz, 1984). Thus, we show detailed considerations on the magnitude of the different error terms in the Supplement.

| **Algorithm 1** Extended Kalman Filter (EKF) | **Algorithm 2** Fixed Interval Kalman Smoother (FIKS) |
|---|---|
| **Initial state:** Expectation $X^{0\|0}$ and covariance $\Gamma^{0\|0}$ | **Initialization:** Run Algorithm 1, store all variables |
| **for** $k = 1, \dots, l$ | **for** $k = l - 1, \dots, 1$ |
| Prediction: expectation and covariance | Backward gain matrix |
| $X^{k\|k-1} = F(X^{k-1\|k-1})$ | $A^k = \Gamma^{k\|k}(\partial F)^T(\Gamma^{k+1\|k})^{-1}$ |
| $\Gamma^{k\|k-1} = \partial F^{k-1}\Gamma^{k-1\|k-1}(\partial F^{k-1})^T + \Gamma_w^{k-1}$ | Smoother expectation and covariance |
| Kalman gain matrix: | $X^{k\|K} = X^{k\|k} + A^k(X^{k+1\|K} - X^{k+1\|k})$ |
| $K^k = \Gamma^{k\|k-1}(H^k)^T(H^k\Gamma^{k\|k-1}(H^k)^T + \Gamma_v^k)^{-1}$ | $\Gamma^{k\|K} = \Gamma^{k\|k} + A^k(\Gamma^{k+1\|K} - \Gamma^{k+1\|k})(A^k)^T$ |
| Measurement update: filter expectation and covariance | **end** |
| $X^{k\|k} = X^{k\|k-1} + K^k(Y^k - H^kX^{k\|k-1})$ | |
| $\Gamma^{k\|k} = (I - K^kH^k)\Gamma^{k\|k-1}$ | |
| **end** | |

**Table 1:** Extended Kalman Filter and Fixed Interval Kalman Smoother algorithms for estimation of the state variables and their variances $X^k$ and $\Gamma^k$.

Considering this structure of the problem, a non-linear extension to the Kalman Filter (Extended Kalman Filter; EKF) is a well suited algorithm for solving the unknown size-distribution and process parameters (Gelb, 1974; Kaipio and

Somersalo, 2005). It is a two stage recursive procedure, where in the first stage the future state and propagation of error is predicted based on the state evolution model (Eq. (3)). In the second stage, the state variable and its covariance are estimated by updating the predicted state variable and covariance. This so-called measurement update accounts for the discrepancy between the realized measurements at time $t_k$ and modelled measurements corresponding to the predicted state variable. This procedure is repeated until the final measurement $k = l$. After finishing the EKF recursions, we utilize a Fixed Interval Kalman Smoother (FIKS), which consists of a backward recursion from a backward gain matrix and smooths the results by backwards recursion from $l$ to 1. The workflow of the EKF and FIKS are illustrated in Table 1 and more details on this algorithm can also be found in Ozon et al. (2020).

**2.3    Adaption to chamber experiments**

The state space model has been adjusted to represent best the evolution of an aerosol system during new particle formation experiments in an atmospheric simulation chamber like CLOUD. For the time evolution of the process parameters, we assume a rather smooth evolution for the nucleation and growth rates, approximated by a second order process (Ozon et al., 2020):

$$J^{k+1} = (r_1 + r_2)J^k - r_1 r_2 J^{k-1} + w_J^k \tag{5}$$

$$g^{k+1} = (r_1 + r_2)g^k - r_1 r_2 g^{k-1} + w_g^k \tag{6}$$

The constants $r_1, r_2$ depend on the characteristic time of change, discretization time and a dampening factor and their definition can be taken from (Ozon et al., 2020) and the corresponding values for our experiments are listed in Table S1 in the Supplement.

In contrast to the growth and formation rates, the loss rates in a chamber experiment do not depend on time, but can be decribed by time independent wall and dilution losses $\lambda = \lambda_{dil} + \lambda_{wall}(d_p)$. These loss rates are well characterized by dedicated wall loss experiments (Stolzenburg et al., 2020) and the dilution rate of the chamber $\lambda_{dil} = Q_{tot}/V_{chamber}$, where $Q_{tot}$ is the total flow rate to the chamber to maintain constant pressure, and $V_{chamber}$ is the chamber volume. The time evolution is described by a random walk with a small stochastic noise term $w_\lambda^k$, and the expectation of the initial state (see Table S1 in the Supplement) is set to the experimentally determined value with a standard deviation of $\pm 10\%$:

$$\lambda^{k+1} = \lambda^k + w_\lambda^k \tag{7}$$

For fully defining the problem (Eq. (3)-(4)), an estimate of the covariance matrices corresponding to the error terms $v^k$ and $w^k$ is needed. The definition of the covariance matrices corresponding to the state noise $w^k$ on the size-distribution evolution $w_N^k$, the growth rate $w_g^k$, the formation rate $w_J^k$ and the wall loss rate $w_\lambda^k$ follow the consideration of Ozon et al. (2020). The covariances of the wall losses, the growth rate and the size-distribution are dominantly diagonal with some additional off-diagonal terms in order to account for a correlation in size. The formulation given by Ozon et al. (2020) was slightly altered to give a stronger correlation between the closest size bins due to the sparser size-resolution of the DMA-train compared to the simulated SMPS system (values for the different experiments are given in Table S1 in the Supplement):

$$\Gamma_{N/g/\lambda}^k(i,j) = \sigma_{i,N/\lambda/g}\sigma_{j,N/\lambda/g} \, \exp\left(-\left(\frac{i-j}{\delta_{N/\lambda/g}}\right)^{a_{N/\lambda/g}}\right) \tag{8}$$

For the size-distribution evolution, we find that $\sigma_{i,N}^2 = (\delta^k)^2 \text{Var}(W_i^k)$, with $\delta^k$ the discretization time-step and $W_i^k$ the error of the discretization of the size-distribution evolution. A detailed derivation of $W_i^k$ is given in the Supplement.

The modelling error of the observation model and the measurement noise both contributing to $v^k$ are assumed to be mutually independent. For this reason, the covariance of the error term $\Gamma_v^k$ in the measurement model is written as a sum

of the covariances of these two random variables, i.e. $\Gamma_v^k = \Gamma_{mod}^k + \Gamma_y^k$. For a detailed derivation we refer to the
Supplement, where we also show that the discretization error is negligible compared to model and measurement error if
a fine enough size discretization is chosen. We approximate $\Gamma_v^k$ with uncorrelated processes, and hence the covariance
matrices are of the diagonal form. For the measurement error, the variance is given by Poisson counting statistics in the
case of a single-particle counting detector such as a condensational particle counter (CPC):
$$\Gamma_y^k(i,i) = y^k(i) \tag{9}$$
For the model uncertainty, we assume the variance of the kernel $\mathrm{Var}(\Delta H_{i,j})$ is composed of an uncertainty proportional
to $H_{i,j}$ (for example due to an offset in the absolute calibration of the instrument) and a shifting size information error (for
example discrepancy between set and actual classified size in a mobility spectrometer). It can then be formulated as
(detailed values for the experiments under investigation can be found in Table S1 in the Supplement):
$$\Gamma_{mod}^k(i,i) = \sum_{j=1}^{q} \left( n(d_j)\Delta_j \right)^2 \mathrm{Var}(\Delta H_{i,j}) \tag{10}$$

## 3 Experimental methods

We use experimental data from the CERN CLOUD experiment (Duplissy et al., 2016; Kirkby et al., 2011) where we
measured particle size-distributions in the sub-10 nm range with a DMA-train (Stolzenburg et al., 2017). The raw data
obtained from the DMA-train is used as input for the analysis of three different sets of experiments performed in the
atmospheric simulation chamber. The DMA-train instrument kernels are also used for modelling an instrument response
to simulated size-distribution data in order to verify the general performance of the FIKS to DMA-train like data.

### 3.1 DMA-train

The DMA-train is an electrical mobility spectrometer, specifically designed to measure sub-10 nm size-distributions
(Stolzenburg et al., 2017). Six identical DMAs are applied in parallel i.e. they sample through the same inlet. They are
set to six distinct but fixed voltages and hence classified sizes. The charging state of the aerosol is pre-conditioned in two
TSI Inc. Advanced Aerosol Neutralizers (Model 3088), each supplying three DMAs at 5.5 litre per minute (lpm) total
flow. We use the Wiedensohler approximation (Wiedensohler, 1988) to describe the steady-state charge distribution at
the DMA inlets. Kallinger and Szymanski (2015) showed that for the used neutralizers the steady-state charge distribution
is still achieved for flow rates up to 5 lpm and we assume that this holds true for 5.5 lpm flow, too. After size classification,
the aerosol is detected in condensation particle counters. Four channels are equipped with TSI Inc. Model 3776 ultrafine
CPCs for detection of aerosols down to 2.5 nm. They were operated at reduced temperature settings in order to increase
the detection efficiency of the smallest particles, achieving a 50% detection efficiency for particles as small as 2 nm
(Wlasits et al., 2020). Two channels of the DMA-train were operated with particle counters specifically designed for sub-
2 nm particle detection using diethylene glycol (DEG), an Airmodus Ltd. particle size magnifier (Model A10, PSM) and
a TSI Inc. nano-Enhancer (Model 3777). Each is used as a booster stage to activate the particles, which are subsequently
detected by either an Airmodus Ltd. CPC (Model A20, for the PSM) or a TSI Inc. CPC (Model 3772, for the nano-
Enhancer). Both channels have a higher aerosol flow rate of 2.5 lpm resulting in a broader transfer function and higher
transmission at the DMA compared to the 1.5 lpm sample flow in the other four channels. The sheath flow at the DMAs
is kept constant at 15 lpm for all six channels.
The constant sampling at fixed sizes allows for either a higher time-resolution at large aerosol number concentrations or
a higher sensitivity towards low number concentrations due to longer signal averaging times compared to a scanning or
stepping differential mobility spectrometer. To increase the number of measured particle sizes, one DMA is still operated
in an alternating mode, switching between 6.2 and 8 nm every ten seconds. The other DMAs are set to classify particles
of 4.3, 3.2, 2.55, 2.2 and 1.8 nm. The instrument kernels are obtained from calibration experiments, where we use the
DMA transfer function and sampling loss characterization from Stolzenburg et al. (2017), the CPC activation efficiencies
from Wlasits et al. (2020) and the charging efficiency was tested to follow the Wiedensohler approximation in Tauber et
al. (2020). The kernel functions for all seven classified sizes are shown in Fig. 1 for an instrument averaging time of 120
seconds, sulphuric acid-like test particles (using the Ammonium Sulfate detection efficiencies from Wlasits et al. (2020))
and including the detector flow rates of each condensation particle counter. Therefore, the kernels can be used to convert
raw particle counts at the detecting CPCs into a particle size-distribution (within an inverse problem) and vice-versa. Note
that for different chemical composition of the input particles, the CPC response might be different. Therefore, the kernels
used for analysing experiments where particles were formed from oxidized organics (Kirkby et al., 2016) or from iodic
acid (He et al., 2021) are different and approximated by the calibration curves for oxidized beta-caryophyllene and sodium
chloride from Wlasits et al. (2020).

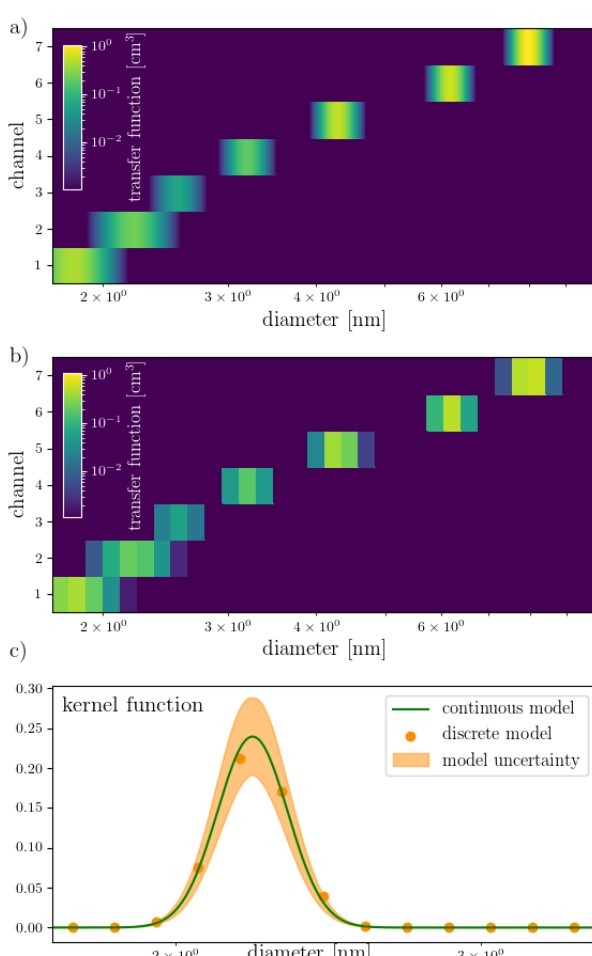


**Figure 1:** Kernel functions of the DMA-train when classifying sulphuric acid particles at a signal averaging time of 120 seconds. (a)
shows the continuous form of the transfer function (color code) of the seven DMA-train channels (y-axis), inferred from instrument
calibrations (Stolzenburg et al., 2017). (b) shows the discretization into 32 size bins from 1.7 to 10 nm used in the Kalman smoother
(c) shows the comparison between continuous form and used discretization for channel 2 with centroid diameter at 2.2 nm, together
with the model uncertainty, which is used for the error estimate (see Table S1 in the Supplement).

## 3.2 CLOUD

We use experimental data from the DMA-train measuring new particle formation in nucleation experiments at the CERN CLOUD chamber. The 26.1 m$^3$ stainless-steel chamber provides a high-purity, temperature-controlled environment in order to perform experiments under atmospherically relevant conditions, where trace gases can be added precisely at pptv (parts per trillion by volume) level and sunlight can be simulated by UV-illumination of the chamber. We use three different sets of experiments, of varying chemical composition, in order to demonstrate the performance of the FIKS in reconstructing formation and growth rates. See also Kirkby et al. (2011) and Duplissy et al. (2016).

First, a nucleation experiment using sulphuric acid and ammonia was performed at 5℃ by adding SO$_2$ and O$_3$ to the chamber and through the photo-dissociation of ozone, the formation of OH radicals and sulphuric acid was induced which resulted in strong new particle formation (Stolzenburg et al., 2020). Second, we performed nucleation and growth experiments at 5℃ using oxidized organics from dark (i.e. no UV-illumination) ozonolysis of alpha-pinene (Kirkby et al., 2016; Stolzenburg et al., 2018). Both experiments resulted in moderate new particle formation rate and thus, in the Kalman smoother, we used DMA-train data that was averaged over 120 seconds time intervals. Third, we studied nucleation from iodine oxides at 10℃ (He et al., 2021), which resulted in high particle formation rates and fast growth. For the third experiment, we reduced the DMA-train averaging time down to 20 seconds, while keeping high counting statistics over the averaging interval.

## 3.3 PSM derived formation rates

Particle formation rates ($J_{dp}$) are calculated from the time derivative of the total particle number concentration larger than 1.7 nm following the method introduced in Dada et al. (2020). The particle number size distribution is measured with the particle size magnifier (PSM) coupled with a condensation particle counter (1.5 – 2.5 nm), a TSI nano-SMPS (3 – 65 nm) and home-built long-SMPS (10 – 800 nm). The formation rates are corrected for the size dependent wall and coagulation losses. Additionally, since the chamber is operated in continuous flow mode, the particle concentrations are corrected for dilution losses. For more information on the PSM derived formation rates, see Dada et al. (2020). The uncertainty on $J_{1.7}$ was assumed to be 30% for the CLOUD chamber derived from the repetition of the same experiment. A procedure as described in Dada et al. (2020) using propagation of error in the concentration measurement, dilution, coagulation and wall losses as well as the error on the time-derivative of the total particle concentration within a Monte-Carlo simulation could be used if such repetition experiments were not available. It needs to be noted, that for comparison of the formation rate value at the arbitrary minimum detectable size $d_{\min} = 1.7$ nm with a system inherent nucleation rate at the critical cluster size, additional sophisticated approaches might be necessary (Kürten et al., 2015).

## 3.4 Growth rates using INSIDE

In order to compare the particle growth rates derived by Kalman smoothing, we use the size- and time-dependent growth rate analysis tool INSIDE (Pichelstorfer et al., 2018). It uses input particle size-distributions at time $t_1$ in order to simulate the known aerosol dynamics (coagulation, wall losses and dilution) until a time $t_2$ (typically separated by one measurement cycle of an instrument, i.e. the 120 seconds averaging time mentioned above). At $t_2$, the simulated aerosol size-distribution is compared to the measured size-distribution and by evaluating the general dynamics equation above a

certain diameter $d_{\text{eval}}$ the growth term $\left.\dfrac{\mathrm{d}d_p}{\mathrm{d}t}\right|_{d_{\text{eval}}} = \dfrac{\left.\dfrac{\mathrm{d}N_\infty}{\mathrm{d}t}\right|_{d_{\text{eval}}}^{\infty} - \left.\dfrac{\mathrm{d}N_\infty^{sim}}{\mathrm{d}t}\right|_{d_{\text{eval}}}^{\infty}}{n(d_p,t)|_{d_{\text{eval}}}}$ can be computed. Evaluating this equation for

several evaluation diameters $d_{\text{eval}}$ and at all measurement times $t_i$, this results in time- and size-resolved growth rates from a size-distribution measurement. This is distinct from most others, integrative growth rate analysis approaches, which can only derive one growth rate value for a specified size-interval in a single run (Dada et al., 2020; Kulmala et al., 2012; Lehtipalo et al., 2014; Paasonen et al., 2018). Last it should be noted that for INSIDE, compared to the Kalman smoothing, each time step is analysed individually and the growth rate analysis framework relies on already inverted size distributions. For this inversion only a simple point-by-point inversion procedure is used for the DMA-train data of this work, assuming narrow DMA transfer functions and little variation of the size-distribution across it (Stolzenburg and McMurry, 2008). Moreover, the INSIDE method does not provide an uncertainty estimate on the growth rate calculation and hence the Kalman smoothing will provide valuable insights on the uncertainty related to growth rate measurements.

## 4 Results: Simulation and Experimental

### 4.1 Numerical simulation test

First, we simulated a data set representing a typical nucleation experiment performed in an atmospheric simulation chamber like CLOUD and modelled the DMA-train response according to the above Kernel functions and then applied the FIKS to this synthetic dataset. We used the same framework as in Ozon et al. (2020) to simulate a nucleation experiment with formation rate at 1 nm $J_{1.0} = 5.25\ \text{cm}^{-3}\ \text{s}^{-1}$, and size-independent growth rate $GR = 2.5\ \text{nm}\ \text{h}^{-1}$ and the loss rates equal to the CERN CLOUD experiment (Stolzenburg et al., 2020). The evolution of the simulated size-distribution is shown in Fig. 2a. The measurement data $y^k$ are then simulated using the kernel functions from Fig. 1a and altered with a Poisson-distributed random counting error. The FIKS is then applied to the measurement data with the input parameters given in Table S1 in the Supplement. We use a resolution of 32 bins from 1.7 to 10 nm for the FIKS to keep the computational effort low. We tested also 16 to 64 size discretization bins, but higher resolution required additional adjustments in the size-correlation of the covariance given in Eq. (8), which would result in significant differences compared to the original work of Ozon et al. (2020) without providing significantly more accuracy. Figure 2 also shows the Kalman smoother estimates for the size-distribution, growth rate and formation rate at 1.7 nm. The reconstructed size-distribution (Fig. 2b) is very similar to the true size-distribution (Fig. 2a), especially taking into account the sparser discretization of the former. Also the estimated growth and nucleation rates agree well with true values of the respective process rates specified in the simulation. Moreover, the uncertainty estimates are feasible: For both quantities, the true values are within the uncertainty limits given by FIKS. The reconstructed size-distribution and especially, growth rate (Fig. 2 b and c, respectively) show some temporal oscillations, which are related to periods for the particle population to grow from the size range visible for one DMA-train channel to the next one. That is, the oscillation is a result of an insufficient coverage of the size-range by the DMA-train kernels. The problem could be approached by application of a regularization scheme in the measurement model (Voutilainen and Kaipio, 2001) or by adjusting the kernels improving the overlap, which will be discussed in more detail in Section 4.5. Nevertheless, the overall good retrieval of the simulated size distribution and process rates demonstrates that Kalman smoothing approach is well suited for analysing DMA-train data from chamber nucleation experiments.

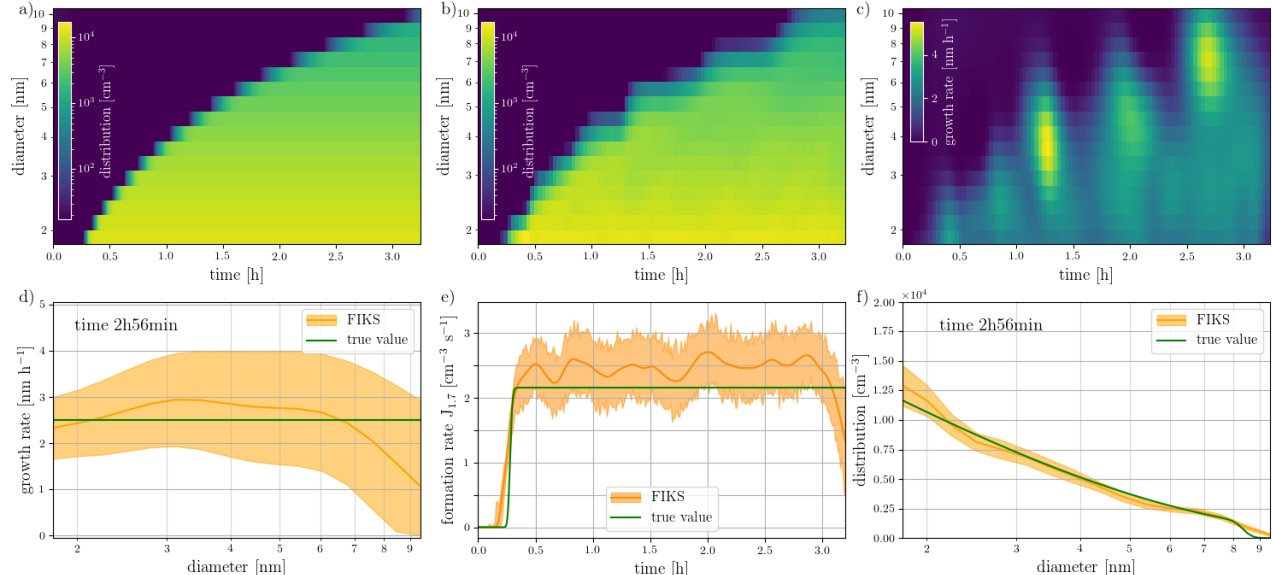

**Figure 2:** Results of the simulated chamber nucleation experiment: (a) The simulated, temporally evolving size distribution of aerosols. (b) The FIKS estimate for the size distribution. This estimate was computed based on the numerically simulated DMA-train data corresponding to the synthetic size distribution shown in (a). (c) The FIKS estimate for the growth rate of particles. (d) The growth rate corresponding to a single instant of time (2h 56min); here, the FIKS estimate and the associated uncertainty limits are plotted together with the true growth rate. (e) The FIKS estimate for the formation rate at 1.7 nm and its uncertainty, and the true formation rate. (f) FIKS estimate, its uncertainty and the true value of the particle size distribution corresponding to a single instant of time (2h 56min). In subfigures d-f, the orange lines represent the FIKS estimates (posterior expectations) and the orange shaded areas illustrate the uncertainties of the associated variables (more specifically, their approximate posterior standard deviation limits). The true values of the quantities are plotted in green.

### 4.2 Sulphuric Acid-Ammonia experiment

We applied the FIKS to experimental data from a sulphuric acid-ammonia nucleation and growth experiment performed at 5℃ at the CERN CLOUD chamber. The raw data measured with the DMA-train were averaged in 120-second time intervals in order to increase the counting statistics per channel and then used as input for BAYROSOL. The details of parameter choices in FIKS are given again in Table S1 in the Supplement. The results of applying the Kalman smoothing to this experimental data are shown in Fig. 3.

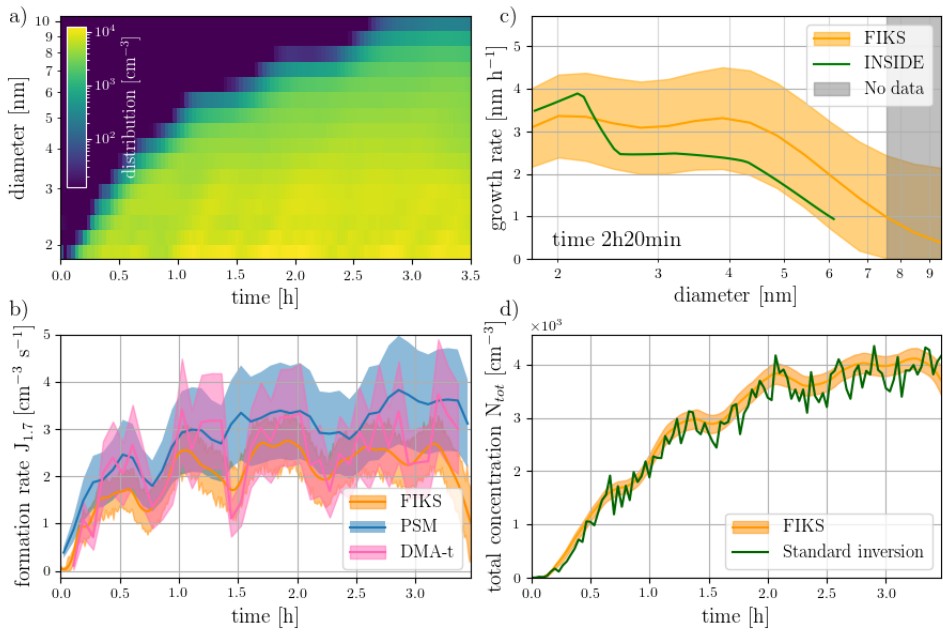

**Figure 3:** Results of an experiment performed at the CERN CLOUD chamber: The DMA-train data was acquired during sulphuric acid-ammonia nucleation and growth processes. Experimental conditions: 5°C, 60% RH, 5 ppb $SO_2$, 120 ppb $O_3$ and 40 ppt $NH_3$. (a) The FIKS estimate for the size distribution. (b) The evolution of the formation rate during the experiment. The orange line and shaded area represent the FIKS estimate and its uncertainty, respectively, while the solid blue line and blue area show the nucleation rate estimate and the associated 30% uncertainty based on a different set of instruments (particle size magnifier, PSM) and the standard approach for inferring nucleation rates at 1.7 nm (Dada et al., 2020). The pink line and shaded area show the standard approach using the data from the DMA-train only. (c) The growth rate corresponding to time 2h20min after the start of the experiment. The FIKS estimate and the associated uncertainty are marked with orange line and shaded area, respectively. The green line represents the growth rate estimate, which is obtained from DMA-train data by traditional inversion (INSIDE method). (d) The total number concentration reconstructed from the standard inversion approach (green line) and FIKS (orange line and shaded area for the uncertainty range).

The size- and time-dependence of the FIKS estimate for the true size-distribution (Fig. 3a) is very smooth, and it is also able to bridge the information gaps between the largest size-distribution channels. This is a significant improvement from the traditional point-by-point inversion, where the data of each DMA-train channel is inverted independently: Here the reconstructed total number concentration is significantly more noisy. The smoother FIKS reconstructed size-distribution and total number concentration (Fig. 3d) still show some small temporal fluctuations. The fluctuations are even more clearly visible in the reconstructed evolution of the formation rate (Fig. 3b). In contrast to the oscillations found in the growth rate for the simulated case, the fluctuations do not only occur when the size-distribution reaches a new DMA-train size channel. Furthermore, the same fluctuations are also recovered when the formation rate is inferred from the PSM and nano-SMPS, i.e. an entirely different set of instrumentation and different type of data-analysis. This result suggests that the fluctuation of the particle formation truly occurs physically in the experiment and is not a reconstruction error caused by instrument noise or a bias caused by the inversion method. The absolute values of the two independent formation rate estimates agree upon a factor of 1.5. Furthermore, large portions of the uncertainty intervals of these two estimates overlap with each other, which is another indicator of the feasibility of FIKS to analysing DMA-train data. In addition, also the inferred growth rates agree within the systematic uncertainties for both approaches (Fig. 3c for one time instant). It is worth noting, however, that both growth rate estimates rely on data from the same instrument and are hence more interdependent than the formation rate estimates. Nevertheless, the good agreement between them corroborates the feasibility of Kalman smoothing to reconstructing nanoparticle growth rates from experimental data.

### 4.3 Alpha-pinene ozonolysis experiment

Third, we applied the FIKS to experimental data obtained by the DMA-train from an alpha-pinene ozonolysis experiment. Besides the different chemical composition of the growing particles (resulting in different assumptions on the DMA-train transfer functions) in comparison to the experimental results used in section 4.2, this experiment is characterized by a different size-dependency of the growth rate, due to the increased condensation of low- and semi-volatile organics with increasing particle size (Simon et al., 2020; Stolzenburg et al., 2018; Tröstl et al., 2016). The formation rate, however, remains rather similar to the sulphuric acid-ammonia experiment under these specific experimental conditions, but the slower initial growth rates result in generally lower produced particle concentrations. The results of applying FIKS to the data from this experiment are shown in Fig. 4. Again, the size-distribution given by FIKS is much smoother than that obtained with a standard inversion procedure, see Heinritzi et al. (2020) and Fig. 4d for the total number concentration evolution retrieved from the standard inversion and the FIKS.

Because in alpha-pinene ozonolysis experiment the nucleation and growth rates were known to be similar to those in the sulphuric acid-ammonia experiment, the parameters of their evolution models were selected as in Section 4.2 (Table S1 in the Supplement). The formation rate is lower by a factor of 2.5 than the one obtained from the PSM. Also the formation rate retrieved with the method from Dada et al. (2020) but using the DMA-train data is significantly higher, but in-between the two estimates. The possible deviation has hence two plausible reasons: The instrumental differences can be caused by

different calibration procedures for the DMA-train and PSM (Dada et al. (2020); direct cross-calibration using NAIS
versus Wlasits et al., (2020) using beta-caryophyllene ozonolysis surrogates). And the methodological differences could
arise from the very low counting statistics in the DMA-train during this experiment compared to the other two, which
will cause the inherent Gaussian assumption of the FIKS to fail. As the deviation in the reconstructed total number
concentration of the DMA-train data using two inversion procedures is only a factor of ~1.3 (Fig. 4d), the formation rate
discrepancies could be largely due to the more difficult and uncertain calibration procedures. Considering the fact that
inter-instrument deviations in sub-10 nm size-distribution measurements can be as large as one order of magnitude
(Kangasluoma et al., 2020), the achieved agreement is remarkable, especially as some fluctuations within the chamber
can again be reconstructed in both approaches. The retrieved growth rates from the FIKS estimate and the INSIDE method
agree remarkably well and both show the increasing growth rates with increasing particle size up to 4.3 nm, an indication
for a strong Kelvin-effect in organic condensation (Stolzenburg et al., 2018; Tröstl et al., 2016). The decreasing Kalman
smoother estimate above 4.3 nm is related to the fact that the FIKS searches a smooth estimate of the growth rate (Eq.
(6)), but at that point (2h30min after experiment start), practically no information from the size-distribution above 4.3 nm
is available, resulting in a slow decrease towards zero. This is due to the strong smoothness a priori used in the FIKS
algorithm (Table S1 in the Supplement). The INSIDE method does not report growth rates from regions with no
information (Pichelstorfer et al., 2018) from the size-distribution and hence the estimate stops at 4.3 nm.

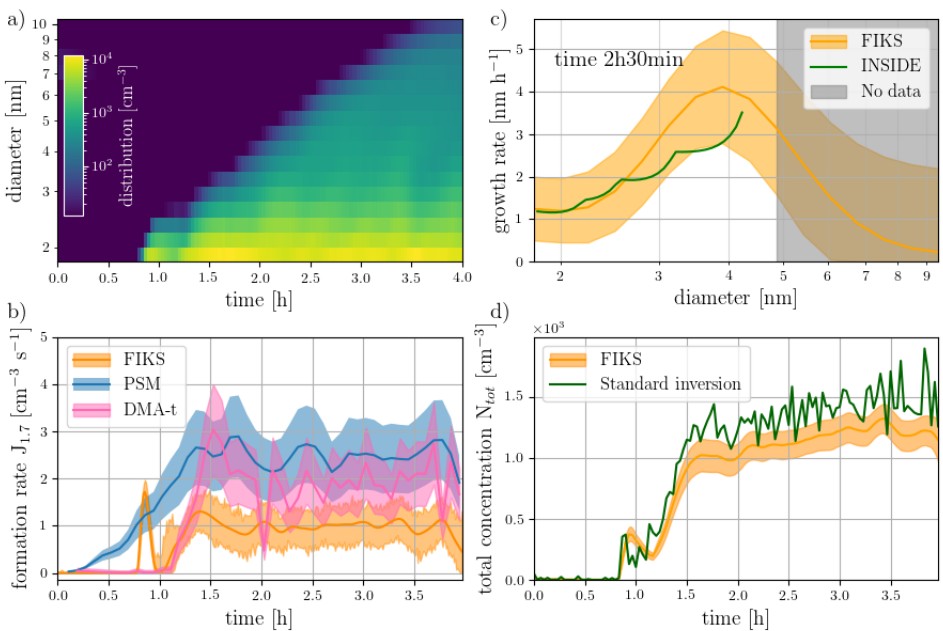


**Figure 4:** Results of an experiment performed at the CERN CLOUD chamber: The DMA-train data was acquired during an alpha-
pinene ozonolysis. Experimental conditions: Experimental conditions: 5°C, 40% RH, 300 ppt alpha-pinene, 40 ppb $O_3$. (a) The FIKS
estimate for the size distribution. (b) The evolution of the formation rate during the experiment. The orange line and shaded area
represent the FIKS estimate and its uncertainty, respectively, while the solid blue line and blue area show the nucleation rate estimate
and the associated 30% uncertainty based on a different set of instruments (particle size magnifier, PSM) and the standard approach
for inferring nucleation rates at 1.7 nm (Dada et al., 2020). The pink line and shaded area show the standard approach using the data
from the DMA-train only. (c) The growth rate corresponding to time 2h30 min after the start of the experiment. The FIKS estimate and
the associated uncertainty are marked with orange line and shaded area, respectively. The green line represents the growth rate estimate,
which is obtained from DMA-train data by traditional inversion (INSIDE method). (d) The total number concentration reconstructed
from the standard inversion approach (green line) and FIKS (orange line and shaded area for the uncertainty range).
**4.4 Iodic Acid experiment**
Finally, we analysed a more dynamic experiment of nucleation and growth from iodic acid (He et al., 2020, 2021). The
experiment is characterized by extremely high nucleation rates, which are two orders of magnitude higher than in the
sulphuric acid-ammonia and alpha-pinene ozonolysis experiments. However, the growth rate is only half an order of
magnitude higher compared to the other two example cases. Figure 5 shows that, in spite of the highly dynamic
experiment, the formation rate recovered by Kalman smoothing agrees with the estimate obtained from the PSM (Fig.
5b). The usage of the DMA-train data with a time resolution of 20 seconds allows for the precise recovery of the spike in
formation rate in the beginning of the experiment. This causes the build-up of a high condensation sink and vapour/cluster
depletion almost shutting off any further nucleation during the continuation of the experiment. The four-minute time
resolution of the data for the calculation of the nucleation rate from the PSM is limited in that respect. The reconstructed
growth rates (Fig. 5c) show again agreement between the FIKS estimate and the INSIDE method, indicating a clear
decreasing trend with size, which is expected for condensation at the kinetic limit if the vapour molecular size is taken
into account (He et al., 2021; Lehtinen and Kulmala, 2003; Nieminen et al., 2010; Stolzenburg et al., 2020). The lower
values towards 1.8 nm could be caused by a biased estimate of the PSM detection efficiency, because neither a calibration
for iodic acid clusters nor for sodium chloride particles (which was used for the other detectors in the DMA-train) was
available. The instabilities in the size-distribution at the smallest sizes and fluctuations of the formation rate are expected
considering the highly dynamic process of this experiment. Overall, the good agreement for the inferred process
parameters of the aerosol general dynamics equation, i.e. the formation and growth rates, is still remarkable. However,
the reconstructed size-distribution from the FIKS estimate shows some discontinuities, especially during the growth above
3 nm. This is because the available instrument information from the DMA-train starts to get very sparse in that size range
given the dynamic processes involved in the iodic acid nucleation and growth. This also causes the overshooting in the
total number concentration using the standard inversion compared to the FIKS result (Fig. 5d), as the larger size channels
cover a very broad range at too high resolution for the linear interpolation of the standard inversion used to obtain $N_{tot}$.
More available size channels (hence more DMAs in the case of the DMA-train) would help to resolve such discontinuities.
We will therefore provide an instrument design recommendation based on simulated data in the next section.

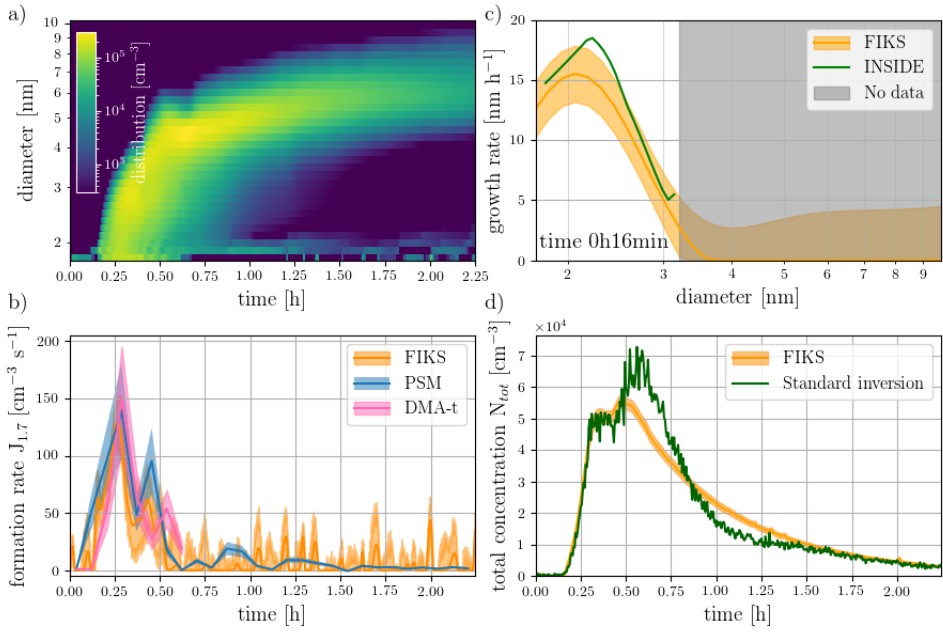


**Figure 5:** Results of an experiment performed at the CERN CLOUD chamber: The DMA-train data was acquired during iodic acid
nucleation and growth processes. Experimental conditions: Experimental conditions: 10℃, 80% RH, 100 ppt $I_2$, 40 ppb $O_3$. (a) The
FIKS estimate for the size distribution. (b) The evolution of the formation rate during the experiment. The orange line and shaded area
represent the FIKS estimate and its uncertainty, respectively, while the solid blue line and blue area show the nucleation rate estimate
and the associated 30% uncertainty based on a different set of instruments (particle size magnifier, PSM) and the standard approach
for inferring nucleation rates at 1.7 nm (Dada et al., 2020). The pink line and shaded area show the standard approach using the data

from the DMA-train only. (c) The growth rate corresponding to time 0h16min min after the start of the experiment. The FIKS estimate and the associated uncertainty are marked with orange line and shaded area, respectively. The green line represents the growth rate estimate, which is obtained from DMA-train data by traditional inversion (INSIDE method). (d) The total number concentration reconstructed from the standard inversion approach (green line) and FIKS (orange line and shaded area for the uncertainty range).

## 4.5    Instrument design recommendation from a signal processing point of view

The size range covered by the seven DMA-train channels was chosen semi-arbitrarily based on some external constraints: the lowest measured centroid diameter was supposed to be as close as possible to 1.7 nm where the formation rate is typically measured for experiments performed at the CLOUD chamber. In order to cover the sub-10 nm range the largest channels was set to 8 nm. The other channel diameters were chosen to yield sufficient coverage of the sub-3 nm range. The width of the transfer functions was fixed by the detector sample flow rates and the 15 lpm critical orifices provided by Grimm Aerosol for the DMA sheath air supply (also the standard flow rate used by the manufacturer for this type of DMA). However, the chosen centroid diameters (and hence fixed voltages at the DMAs) and selected sheath flow rates could easily be altered.

In order to study numerically the effect of choice for the DMA-train channels, we constructed a model corresponding to a channel choice different from that in the previous sections: The kernel for the DMA-train with seven channels having centroid diameters of 1.8, 2.2, 2.8, 3.6, 4.6, 6.1 and 8.0 nm is illustrated in  Fig. 6a. Moreover, channels 3 to 7 have been altered to use a 2.5 lpm sample flow rate (could be achieved by a make-up flow at each CPC) and a reduced sheath flow rate of 10 lpm, only providing a theoretical non-diffusive resolution of ~4, which is significantly lower. However, this permits covering the entire size-range between 1.8 and 8 nm by overlapping channels. We revisited the numerical simulation described in Section 4.1 using DMA-train model corresponding to this configuration. The resulting size-distribution given by FIKS is shown in Fig. 6b. Comparison between the two reconstructions in Fig. 2b and Fig. 6b reveals that the new choice of DMA-channels stabilizes the size-distribution estimate, which is supported by the comparison of the estimated total number concentrations (Fig. 6d). While with the original design the size distribution evolves in step-wise manner when the growing particle mode reaches larger size-channels, the size distribution retrieved with the adjusted kernels is smoother, also temporally, which is especially visible in the evolution of the total number concentration. The improvement is also significant for the retrieval of the growth rate, where the estimate no longer overshoots as illustrated by the snapshot size dependence of the growth rate for a single instant in time, shown in Fig. 6c. It should be noted, that also regularization schemes in the measurement model as proposed by Voutilainen and Kaipio (2001) could provide smoother estimates, but this would need a significant adjustment of the algorithm provided by Ozon et al. (2020) and is hence not implemented here.

The result of this additional numerical study thus demonstrates that in the chosen conditions of the simulation, the reconstruction quality improves when the resolution of individual channels is lowered. This seemingly counterintuitive effect stems from the fact that FIKS estimates do not rely only on the measurements, but are also advised by the GDE model, which makes the problem of optimizing the measurement design a somewhat cumbersome task. A rigorous investigation of the optimal experimental design is out of the scope of this paper, but the above observation is worth noticing -- especially because a lot of recent experimental effort in the sub-10 nm range has been devoted to improving the instrument resolution (Kangasluoma et al., 2020). While on the individual channel level this might reduce systematic uncertainties, as discussed in Kangasluoma et al. (2020), signal processing rather requires a broad coverage of the size distribution than high resolution. However, the ideal instrument would combine both, full coverage of the size distribution but achieved with more, high-resolution channels. For the DMA-train principle, this would require new ideas in

instrument design in order to incorporate more DMAs without the instrument becoming impractically bulky. Extending the measurement size-range above 10 nm would require even more DMAs, but this range is usually well-covered by commercially available instruments, which could easily be added to the FIKS measurement model, facilitating the measurement of experiments where particle grow well beyond 10 nm.

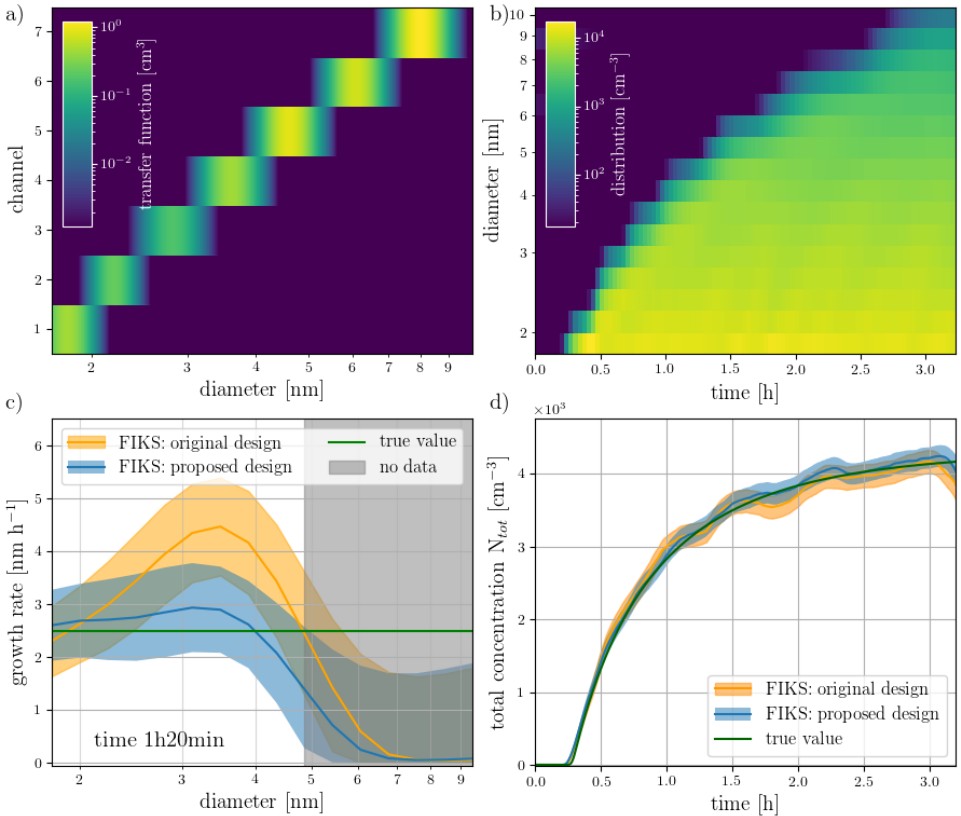

**Figure 6:** Adjusted DMA-train kernel for better signal processing (a), which could be achieved by choosing suitable centroid voltages and altering the resolution of the DMAs through sheath-flow and sample-flow adjustments. (b) shows the reconstructed simulated size-distribution of Fig. 2 using the adjusted kernel. (c) The growth rate corresponding to time 1h20min min after the start of simulation. The orange line and shaded area are the FIKS estimate using the original design of the DMA-train, while the blue line and shaded area indicate the FIKS result from the proposed new design. The green line shows the true value used to create the input size distribution(d) shows the evolution of the total particle number concentration for the input size-distribution (green), the original kernel (orange line and shaded area for the uncertainty) and the proposed design (blue line and shaded area).

## 5    Conclusion

A recently developed methodology (BAYROSOL) applying the Fixed Interval Kalman Smoother (FIKS) to a finite difference solution to the aerosol GDE was used to analyze DMA-train measurements of aerosol dynamics at the CLOUD chamber facility at CERN. The overall aim of this methodology is to estimate unknown aerosol microphysical process rates as well as their uncertainties from size-distribution evolution measurements. In a previous paper, the methodology was shown to be able to predict new particle formation, growth and loss rates from synthetic computer-generated aerosol size distribution evolution data, while here the method has been applied to real experimental data for the first time. Four experimental cases with particle formation and growth were tested: 1) a computer-generated synthetic case, 2) sulphuric acid-ammonia, 3) alpha-pinene ozonolysis, and 4) iodic acid.

The DMA-train was selected for two main reasons: first, the instrument kernel functions are well characterized giving rise to reliable estimation of the instrument uncertainties and second, new particle formation rates have not been estimated directly from DMA-train measurements before. In addition, as the current version of the DMA-train is designed in such

a way that the individual DMAs have a rather narrow collection kernels for the channels, with significant gaps between some of the channels, the FIKS can reconstruct the size distributions from the measured signals in such a way that the distributions are rather smooth and consistent with the GDE.

We compared the growth rates, which with BAYROSOL can be estimated as functions of both size and time, with those obtained by INSIDE, a method applied earlier to CLOUD data, and the agreement was remarkably good for all studied cases. INSIDE is also based on matching the GDE solution to measured size distribution dynamics, however without the capability of estimating uncertainties of the estimations. The FIKS-based estimates for the particle formation rates were compared with those estimated from data obtained by a separate instrument, the Particle Size Magnifier (PSM), based on the rate of change of the total number concentration measured by the instrument corrected by coagulation and wall-losses. Again, the agreement was very good, especially considering the fact that instrument uncertainties are large at the very lowest end of the measured size spectrum. For the iodic-acid case, the FIKS estimate of the formation rate was even able to capture rapidly changing dynamics of the experiment. It was remarkable for all cases that some fluctuations in the formation rates were recovered by both methods independently, indicating that these are physical variations during the experiment

Finally, we utilized the FIKS from an instrument development point of view. Typically, an as-high-as-possible resolution for the different measurement channels has been the aim when measuring nanometer-sized particles. This aim, however, gives rise to gaps in the measured size range, as is the case in the DMA-train studied. Thus we studied whether a better coverage of the size spectrum, but lower resolution of the individual channels would be advantageous for size distribution estimation.

Summarizing, we believe that Bayesian state estimation methods such as FIKS can be very useful in the field of aerosol science in many aspects. As mentioned above, they can be used to fill gaps in measurements in such a way that not only the obtained size distributions but also unknown process rates are consistent with theory describing aerosol size distribution dynamics. In addition, the methodology provides estimations of uncertainties both for size distributions as well as process rates based on uncertainty estimations in the measurements and the used models, which is unfortunately not common when reporting results of aerosol measurements. Finally, it is conceivable that the methodology presented here will be superior to several previous approaches when combining measurement data obtained with several different instruments that operate at different size ranges. This will be a topic of our forthcoming studies.

**Data availability**

The version of the implementation of the estimation method (BAYROSOL1.1)[https://doi.org/10.5281/zenodo.4450492] is available under the MIT Expat License; it is the version used to generate the results discussed in this paper. The package also contains the code used to obtain the results discussed in Ozon et al. 2020. It is possible to generate the simulated data described in section 4.1. The experimental data (DMA-train kernels) and raw data, are also included in the repository.

**Competing interest**

The authors declare no competing interests.

**Acknowledgements**

This publication has been produced within the framework of the EMME-CARE project, which has received funding from the European Union's Horizon 2020 Research and Innovation Programme (under grant agreement no. 856612) and the Government of Cyprus. This research has also received funding from the European Union's Horizon 2020 research and innovation programme under the Marie Sklodowska-Curie grant agreement no. 895875 ("NPF-PANDA") as well as Academy of Finland Project #325647. The sole responsibility of this publication lies with the author. The European Union is not responsible for any use that may be made of the information contained therein.

**Author contributions**

MO, DS, LD analysed the experimental data, MO developed the software, all authors were involved in the scientific discussion and in the writing of the manuscript.

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
