# Peer review of "Aerosol formation and growth rates from chamber experiments using Kalman smoothing"

_Atmospheric Chemistry and Physics, 2021_

## Author Comment (AC1)

**Reviewer replies: Reviewer #1**

We thank the reviewer for her thoughtful comments and will revise our manuscript accordingly. In the following are our point-by-point responses to the reviewer's remarks in such a way that we have listed the reviewer's remarks in blue and our replies in black font and changes to the manuscript in red.

Ozon et al. present the method and analyses the performance of a Bayesian technique using a Kalman filter for calculating aerosol size distributions, growth rates and nucleation rates from a specific instrument (the DMAtrain) in chamber experiments. They also use this analysis to make instrument design recommendations which are of interest beyond this specific instrument and perhaps beyond chamber experiments. I believe that this manuscript may be better suited for Atmos. Meas. Tech., being of primarily technical interest rather than containing general implications for atmospheric science.

The details of the Bayesian algorithm are currently in review at Geosci. Model Dev. (Ozon et al. (2020)). This manuscript focuses on the application to chamber data, which is valuable to the community, and seems very appropriate to keep in a separate paper form the one describing the algorithm. Advancement of techniques aerosol size distribution inversions, and calculation of processes rates, and the explicit calculation of related uncertainties as presented here, is of value of the aerosol science community. Large uncertainties in growth and nucleation rates, and a lack of mathematically rigorous techniques for calculating them has made it hard for the community to compare between different experiments and to fully understand the effects of different environments and emissions on aerosol climatic and health effects. This analysis addresses these problems for the specific case of the DMAtrain on the CLOUD chamber, which is of value in and of itself, but also as an example for those developing techniques for other instruments or environments. The implications for instrument design drawn explicitly from the analysis presented here are of great value to the community and have the potential to influence thinking on design of a range of aerosol instrumentation.

This manuscript relies heavily on the algorithm presented in Ozon et al. (2020), with more limited explanation here. My review is based on the analysis of results presented in this study only, and not the details of the algorithm presented in Ozon et al. (2020). I note than Ozon et al. (2020) is currently in review, and so caution that publication of this work should depend upon successful peer-review of that work which it builds upon.

I have three main concerns with this manuscript regarding the science presented:

- A major claim of this manuscript is that the nucleation rates calculated from this method agree with nucleation rates calculated using a different instrument and method. Figure 4b shows that, within calculated uncertainties, this is not consistently true. The authors mention that disagreement between instruments is often order of magnitude as opposed to the much smaller disagreement shown here. This suggests that this method is a great improvement upon previous methods, which is very valuable to our field, but more rigor and accuracy is required in how the result is described, or in explaining why we might not expect these two results to agree within uncertainties.
- There is a general lack of quantitative comparison between results from different methods

- • I was unable to fully understand some aspects of the method and remain unconvinced about some method choices made in this study. This may be a clarity of presentation issue, or things that need addressing in the analysis itself.

I will address these specifically, and other issues in the line-by-line review below.

Regarding clarify of presentation, I am concerned that this manuscript relies too heavily on other literature, to the extent that it is not possible to fully understand the work presented without constant reference to other literature. Specifically, this manuscript relies heavily on Ozon et al. (2020) to an extent that the methods are not understandable without either having a very intimate knowledge of that paper, or keeping both papers open and constantly going back and forth between them. While it make sense to build on the prior work and not reproduce too much in a new paper, something should be done to make this paper more understandable on its own. I would also like to mention that, while not incorrect, the paper, being aimed at a general atmospheric chem/phys audience in this journal, might be more readable if the authors explained the general idea of Bayesian state estimation problems to a general chemistry audience. Ditto Kalman filters. I also note that this manuscript contains a lot of symbols with definitions placed throughout the text. A table defining all terms somewhere accessible would ease readability.

Thank you for this comment – we agree. First, the paper by Ozon et al. in GMD has now been published (doi: 10.5194/gmd-14-3715-2021). Second, to familiarize the readers somewhat to the philosophy of Bayesian state estimation, we add the following section to the Introduction-section of the manuscript:

"Bayesian state estimation is a general framework for estimating time-dependent variables (state variables) based on (direct or indirect) noisy observations that are collected sequentially during the temporal evolution of the state variables (Gelb, 1979). The state estimation is based on the so-called state-space representation, which consists of the state evolution model and observation model. In this work, the state variables consist of the particle size distribution – the temporal evolution of which is modelled with GDE – and the nucleation, growth and deposition rates which are parameters of the GDE (Ozon et al., 2020). The observation model is the mapping from the size distribution to DMA-train measurements. In Bayesian formulation, both the state variable and the observations are modelled as stochastic processes; their randomness reflects their uncertainty, which decreases, when measurement data is accounted for in the state estimation – formally speaking, this is done by conditioning the state variables with respect to measurement data (realized observations) sequentially. The result of Bayesian state estimation is the posterior probability density which reflects the uncertainty of the state variables after accounting for the measurement data.

A large variety of state estimation schemes exits, and the choice between them depends on 1) the type of the state-space model (linearity, gaussianity, etc), 2) the type of data available when computing an estimate at time t (if data is available up to time $k < t$, the problem is of prediction type, while cases where $k = t$ and $k > t$ are referred to as filtering and smoothing, respectively), and 3) the approximations which are sometimes needed to lower the computational demand of state estimation. In the case of linear and gaussian state-space models, a Bayesian filtering problem can be solved recursively by the well-known Kalman filter algorithm. In non-linear and non-gaussian cases, the rigorous choice is to use so-called particle filters and smoothers (Särkkä, 2013). However, because these MCMC-based estimators are highly time

consuming in large-dimensional cases, approximative methods are often used – such as the extended Kalman filter and smoother adopted in this paper. These recursive algorithms use sequential linearizations to approximate the non-linear models and non-gaussian probability distributions. In the previous work (Ozon et al., 2020), these approximations were shown to lead feasible state estimates for the particle size-distribution and process rates with synthetically generated DMA-data."

In terms of adequate referencing of existing literature, there are a number notable omissions in this manuscript. The discussion of formation and growth rate calculation methods makes no mention made of Kurten et al. (2015), despite this also being developed for the same CLOUD chamber. Another method used frequently on data from the CLOUD chamber is the use of the AeroCLOUD model, as described in the methods section of Kirkby et al. (2016). The work of (Fiebig et al., 2005) has also been missed. Without placing the work presented in the context of these other studies, it is hard to assess its importance and relative merits. Lack of knowledge of some of this literature seems to have led to erroneous statements in the text, which I will point out below.

The main motivation of our manuscript is not to compare the best estimate results of our method to all possible ways to estimate process rates. Instead, the motivation of the comparisons presented was to show that the estimations are reasonably close to the estimation methods used earlier by the authors as well as highlight the fact that as a result we get probability density functions (estimates for uncertainties) in addition to the best estimates. However, we agree with the reviewer that the techniques she mention have been important developments when considering CLOUD data analysis, and mention these in the Introduction section of the revised manuscript. In the Figure below we show that the nucleation rates estimated by the method in Dada et al. with the PSM agree well with estimates from AeroCLOUD. In the most recent campaigns, the AeroCLOUD method was thus discarded and the direct measurement at 1.7 nm was preferred. Similar agreement has been shown for the method of Kürten et al., 2015. However, we cannot include a comparison for the shown datasets with AeroCLOUD or the method by Kürten et al. (2014) as this group of authors has no access to the code and the comparison would be well beyond the scope of the manuscript. Regarding potentially erroneous statements, we reply to them in the more detailed comments below.

[Figure]

*Supporting Figure 1: Comparison of the nucleation rates estimated by AeroCLOUD and those from Dada et al. (2020) for a*

*different set of experiments.*

Lines 18-19: Agreement between growth and nucleation rates described as "remarkably good" and "matched .. well". A quantitative description of the agreement is needed.

We chose this wording because for the growth rate, the estimate given by the previous INSIDE method falls within the uncertainty range (one standard deviation) of the Kalman smoother estimate for all particle sizes in the sulphuric acid-ammonia and alpha-pinene ozonolysis experiments, and for all but a very small size range in the iodic acid experiment. Note that the INSIDE method matches aerosol dynamics only to the 'pointwise' data given by the DMA train of the size distribution while the Kalman smoother reconstructs the whole size distribution also in the measurement gaps. Furthermore, even if the match for the nucleation rates is not quite as good, especially for the alpha-pinene ozonolysis experiment (roughly a factor of two difference in rates), we consider it still quite remarkable as we compare against PSM measurements where the uncertainties at the smallest sizes can be very large and are not taken rigorously into account in the simple estimation method (Dada et al., 2020).

Still, we remove the word remarkable and change the wording of the abstract in the following way in the abstract:

"The agreement in the growth rates was very good for all studied cases: estimations with an earlier method fell within the uncertainty limits of the Kalman smoother results. The formation rates matched also well, within roughly a factor of two in all cases, which can be considered very good considering the fact that they were estimated from data given by two different instruments, the other being the Particle Size magnifier (PSM), which is known to have large uncertainties close to its detection limit."

Line 19: Formation rate should be independent of the instrumentation used, so I'm not sure that the qualification of "especially given the fact that they were calculated from different instruments" is meaningful, at least not without a thorough explanation of why one might expect them to be different.

See answer to previous comment. Measuring particle size and number concentration close to nucleation size is very difficult, and it is known that especially devices that rely on activating particles for growth, such as the PSM, can have significant uncertainties (Kangasluoma, 2020). This, together with the simple 'traditional' method for estimating formation rates is so different to DMA train & GDE & Kalman smoothing that differences in results are no surprise.

Line 27: ``their concentration'' it is unclear what that is referring to, I suggest being explicit.

changed to: "Aerosol number concentration"

Line 68 – The Dada et al. formation rate calculation method is mentioned here but not included in the earlier paragraph describing the different methodologies for calculating formation rates. Why?

now added where Kulmala et al. (2012) is mentioned.

The description of why the DMAtrain is used is a little puzzling. Were other instruments available which did not meet the criteria laid out? This section needs clarification in terms of motive along with more specific clarification as follows:

To motivate better the use of the DMA train, we modified the end of the Introduction to read: "We chose the DMA-train measurements for three main reasons: first, the high time resolution makes it an ideal instrument for nucleation studies due to a more accurate estimate of dN/dt. Second, the collection efficiencies of the channels have been carefully characterized (Stolzenburg et al., 2017; Wlasits et al., 2020) and yield higher sensitivities to low particle number concentrations (Kangasluoma et al., 2020), which are often faced in atmospherically relevant nucleation studies, and third, the DMA train is, at the same time, interesting and challenging instrument for detailed data analysis because of the gaps in the measured size spectrum."

Line 70 – high time resolution – what is the time resolution of the measurement and how does this relate to rates of change of key observables in nucleation events in the chamber?

High time-resolution is required for the most accurate estimate of dN/dt which is related to the formation rate (Kulmala et al., 2012). Furthermore, the dynamics of the iodic acid nucleation evolve within 15 min. Capturing such quick changes most accurately was one of the design arguments of the DMA-train (see Stolzenburg et al., 2017).

Line 71 – have the collection efficiencies for other candidate instruments not been as carefully characterized?

Yes, of course. Some of them are, however, not directly available for commercial instruments. The DMA train was just a convenient choice because of the above mentioned reasons. Especially, it has been shown in Stolzenburg et al. (2018), Heinritzi et al. (2020) and Stolzenburg et al. (2020), that the DMA-train is more sensitive to low particle number concentrations in the sub-10 nm range compared to e.g. the TSI nano-SMPS, when measuring CLOUD nucleation experiments.

Line 73 – it would be helpful to explain why it is advantageous to have a new instrument to calculate formation rates from instead of optimizing methods for instruments that have been used previously

We thank the reviewer for pointing us towards a more thorough description of why the DMA-train is used. As mentioned above, we modified the sentence in the revised version pointing towards the facts that an accurate estimate of dN/dt is beneficial for any nucleation rate estimate, and hence the time-resolution of an instrument can be crucial, especially in fast evolving experiments like the iodic acid nucleation where the dynamics evolve dramatically within 15 min. We also point out that the DMA-train achieves higher sensitivities for small particles close to the critical size at around 1.8 nm than most other available instruments, details can be found in Kangasluoma et al. (2020). Last, we are convinced that the newest and best instruments should be used if available!

Line 76 – there are also other quantities of interest e.g. chemical composition of cluster, formation rate at larger sizes, coagulation rates … might be better to rephrase this sentence.

We changed "The quantities of interest" to "Typical quantities of interest".

 – "often done by analysis of time evolution of retrieved particle size distributions" this needs references.

We added the citation of Dada et al. (2020) here, which as a Nature Protocol provides some standards for this type of retrieval of formation rates. It includes many references to studies where such methods have been applied. We therefore think that a citation of Dada et al. (2020) is enough.

 – states coagulation can be neglected in certain cases with low particle concentrations – in the context of this study this seems misleading. Coagulation has been explicitly shown to matter in the context of chamber new particle formation experiments Kurten et al., 2015, which can still be considered to be low concentration environments.

In this general context of the description of the general dynamic equation our statement was not targeted explicitly at the presented arguments. We wanted to express that coagulation above the minimum detectable size is negligible if number concentrations are low. We thus added "above $d_{min}$" to clarify this. We agree with the reviewer that the study of Kürten et al. (2015) should be mentioned and hence we added a statement to clarify the difference between formation and nucleation rate: "It needs to be noted, that for comparison of the formation rate value at the arbitrary minimum detectable size $d_{min} = 1.7$ nm with a system inherent nucleation rate at the critical cluster size, additional sophisticated approaches might be necessary (Kürten et al., 2015).". The terminology formation rate was anyways already used throughout the manuscript.

 – "size space" is this particle diameter? Needs to be explicit.

We modified: "After discretization of the problem in size space, i.e. in terms of particle diameter"

 "The incorporation of positivity constraints for the process rates and the aforementioned second order models require minor modifications in the definition of the state variable $X^k$ – it is not clear enough what the aforementioned second order models are, and the modification needed should be shown explicitly. I can understand why some of the methodology refers to Ozon et al. (2020), but for instances such as this it makes reading the paper and understanding the method too time consuming and difficult. Readability in general would be greatly improved if the method could be better understood from this paper without need to refer to other literature as much.

We have made several edits to the manuscript to make the paper more stand-alone. Regarding the sentence in lines 117-118; the reason for originally leaving out the descriptions of the second-order models and positivity constraints was that including them could make the notations of the paper difficult to follow. In the revised manuscript, we will explain the inclusion of the second-order models and positivity constraints to clarify their effects on the state-estimation. At the same time, we will still try to avoid explaining same details as in paper Ozon et al. (2020) to keep the notations simpler and to avoid lengthening the paper too much. The following text was added:
"We note that the above description of the state-space model (3)-(4) is slightly simplified for the sake of notational convenience. Namely, two additional features – both described in detail by Ozon et al. (2020) – are included in the model: First, we assume that the process rates are positive quantities and incorporate this positivity constraint into the evolution model by reparametrizing these quantities in the model. For example, for the nucleation rate $J^k$, we write $J^k =$

$\frac{1}{\alpha}\ln(1 + e^{\alpha\xi_J^k})$, where $\xi_J^k$ is an unconstrained random variable and $\alpha$ is a scaling constant. Respective parametrizations are written for $g_i^k$ and $\lambda_i^k$. Secondly, as noted above, second order models are written for rates $J^k$ and $g^k$. More specifically, we consider the respective state parameters $\xi_J^k$, $\xi_g^k$ as second order Markov processes; for example $\xi_J^k = \psi_1\xi_J^{k-1} + \psi_2\xi_J^{k-2} + \eta$, where $\psi_1$ and $\psi_2$ are model parameters and $\eta$ is Gaussian state noise. The second order models are written, because they promote temporal smoothness of the processes. When the positivity constraint and the second order models are included in the model, the state variable $X^k$ in the state-space model (3)-(4) is rewritten in the form $X^k = \begin{bmatrix} N^k & \xi_g^k\ \xi_g^{k-1} & \lambda^k & \xi_J^k\ \xi_J^{k-1} \end{bmatrix}^T$, and at each time step the above logarithmic functions are used for mapping the unconstrained variables $\xi_J^k$, $\xi_g^k$ and $\xi_\lambda^k$ to respective quantities $J^k$, $g_i^k$ and $\lambda_i^k$. For the details on the above modifications as well as discretization of the GDE model, we refer to Ozon et al. (2020)."

Line 149 - loss rates were defined as lambda above, but lambda doesn't appear in eq. 5 and 6. By loss rates is the author referring to $-r_1 r_2 J^{k-1}$ ? This needs to be better defined

Equations 5 and 6 refer to the particle formation and growth rates, respectively. In contrast, the loss rates do not depend on time – they only depend on size. This size dependence is typically well characterized by dedicated wall loss experiments, as is the case for CLOUD, and as explained after equations 5 and 6, see e.g. Stolzenburg et al. (2020). In our approach, lambda in the loss term is described by a random walk with expectation set to the experimentally determined values and a small stochastic noise term (equation 7). We clarify by modifying after equation 6:
"In contrast to the growth and formation rates, the loss rates in a chamber experiment do not depend on time…." and made a new paragraph to show that Eq. (5) and (6) have nothing to do with the loss rate.

Line 150 - $Q_{tot}$ and $V_{chamber}$ need defining. Also where does this dilution time come into the calculations presented here?

we added: "In contrast to the growth and formation rates, the loss rates in a chamber experiment do not depend on time, but can be decribed by time independent wall and dilution losses $\lambda = \lambda_{dil} + \lambda_{wall}(d_p)$." in order to define the loss rates properly and refer the reader to Table S1 in the Supplement where the initial assumptions are specified. Moreover, we added "(…) where $Q_{tot}$ is the total flow rate to the chamber to maintain constant pressure, and $V_{chamber}$ is the chamber volume", to define the variables properly.

Line 159 – what is the justification for stronger correlation between closest size bins in this method vs Ozon et al 2020?

Strengthening the correlations between values in neighbouring bins implies an assumption of increasing the smoothness of the corresponding quantities with respect to particle size. Rigorously, the choice of smoothness should depend purely on the prior knowledge available on the quantities. On the other hand, in Ozon et al. (2020), the number of measurement channels was very large (more than 100 overlapping channels in the size range of interest). Thus, in Ozon et al. (2020) the state estimates were feasible even though we made weaker assumption on the smoothness with respect to particle size. In the present paper, data from only seven measurement channels is used, a bit stronger smoothness assumption is needed. We clarified this by adding just before equation 8: "The formulation given by Ozon et al. (2020) was slightly altered to

give a stronger correlation between the closest size bins due to the sparser size-resolution of the DMA-train compared to the simulated SMPS system".

Line 179 – only sub 10nm size distribution measurements are mentioned. What about particles that grow > 10nm? These will surely contribute to coagulation sinks and therefore need to be accounted for?

The reviewer is, of course, correct in stating that larger particles will contribute to the sinks. Here, however, all presented data start with a clean chamber and stop when the distribution grows past 10 nm). The Kalman Filter method is flexible with respect to adding any additional size distribution data as soon as another instrument would be included that measures larger particles. We added a sentence to clarify this in Section 4.5 when we discuss instrument design principles: "Extending the measurement size-range above 10 nm would require even more DMAs, but this range is usually well-covered by commercially available instruments, which could easily be added to the FIKS measurement model, facilitating the measurement of experiments where particles grow well beyond 10 nm."

Line 190 – steady state charge distribution achieved for flow rate up to 5 lmp. Flow rates used are 5.5. lpm. How does this affect the collection efficiency? Some justification is needed for going outside of the steady state flow range.

We agree with the reviewer that steady-state for 5.5 lpm was never experimentally verified, but we simply assumed that this still holds true. The overall uncertainties in the Wiedensohler approximation for the sub-5 nm range by far outweigh the uncertainty coming from the 10% deviation of the flow rate. To our knowledge all currently used sub-5 nm mobility spectrometers rely on an extrapolation of the charging efficiency, as even the study by Kallinger and Szymanski only measured down to 5 nm. We added a clarifying subsentence "and we assume that this holds true for 5.5 lpm flow, too." that we rely in this assumption in section 3.1

Line 194-199 -- description of 2 stage CPCs is a bit confusing. Are two "boosters" used on each channel? Is it a different booster for the different channels? If so why?

In section 3.1. we explicitly state that the 3777 is used together with the 3772 and the PSM with the Airmodus. However, to clarify this, we slightly adjust our wording here by stating in section 3.1: "Two channels of the DMA-train were operated with particle counters specifically designed for sub-2nm particle detection using diethylene glycol (DEG), an Airmodus Ltd. Particle size magnifier (Model A10, PSM) and a TSI Inc. nano-Enhancer (Model 3777). Each is used as a booster stage to activate the particles….". The DMA-train has been used in a variety of studies (e.g. Stolzenburg et al., 2018; Lehtipalo et al., 2018; Yan et al., 2020, Stolzenburg et al., 2020, Heinritzi et al, 2020, Kangasluoma et al., 2020; Brilke et al., 2020; He et al., 2021), where its reliability has been demonstrated. Further, the DMA-train is an instrument of value ~500 k€. and some choices for the particle counters had to be made out of availability constraints, which we do not want to elaborate on in this manuscript.

Fig 1 c. legend refers to fine and coarse models, but caption refers to continuous and discretized forms of the kernel functions. Are these the same? I find this reference to models more confusing than kernel functions, but am aware that could be my own bias from the literature I'm familiar with. I do highly recommend the authors stick to a single

terminology to describe the kernel and make sure it is well defined for non-experts. Also is the uncertainty for the continuous function, or both? This should be clarified.

We agree that the terminology should be consistent and adjusted the Figure legend. The uncertainty applies to the mathematical model. We assume some discrepancy between the mathematical model and the true kernel function of the device. We choose to plot only the one related to the finely discretized model for the sake of clarity in the figure.

Line 242 - "chamber is operated in continuous mode" did you mean "continuous flow mode"?

Yes. Adjusted.

Line 244-246 - "assumed" J uncertainty – is this assumed or calculated as described in the rest of the sentence. Working for this uncertainty calculation needs to be shown here or in supplementary material as it is not reproducible with the current level of detail.

We agree with the reviewer that this formulation was not correct. The uncertainty of the formation rate calculation using the PSM method is an empirical uncertainty derived from the repetition of the same experiment over several different experimental runs, i.e. years. The resulting variation in formation rate was found to be within 30 %. A procedure as described in Dada et al. (2020) could be used if such information would not be available, however, we think that the former method is less cumbersome and yields a solid estimate of the uncertainty already. We thus adjusted the sentence to: "(…) derived from the repetition of the same experiment. A procedure as described in Dada et al. (2020) using propagation of error in the concentration measurement, dilution, coagulation and wall losses as well as the error on the time-derivative of the total particle concentration within a Monte-Carlo simulation could be used if such repetition experiments were not available."

Line 255 - "which is distinct from most others" -- does this mean that most growth rate calculation methods do not include size and time dependency? This must be clearer and needs references.

Yes, exactly. To clarify this, we added the following sub-sentence and references to section 3.4:
"which can only derive one growth rate value for a specified size-interval in a single run (Dada et al., 2020; Kulmala et al., 2012; Lehtipalo et al., 2014; Paasonen et al., 2018)"

Line 256 - "However, compared to the Kalman smoothing, each time step is analysed individually and the analysis framework relies on already inverted size distributions, where a point-by-point inversion procedure is used for the DMA train data of this work (Stolzenburg and McMurry, 2008)." This sentence is confusing and I don't understand what it means.

We adjusted the sentences to make them better understandable:

"However, compared to the Kalman smoothing, by which both the data and the time-evolution model are used to reconstruct the size distribution, each time step is analysed independently and the analysis framework relies on already inverted size distributions. The size distribution is inverted from the DMA train data using a simple inversion procedure, namely a point-by-point inversion, which relies on a singly charged particle assumption. It is used for the DMA train data of this work (Stolzenburg and McMurry, 2008)."

Line 260 – if INSIDE cannot provide uncertainty on the growth rate, I would argue that another method is not so much needed to "verify" the result as the authors state here, but to provide a growth rate with calculated uncertainty, and the meaning of a rate with no uncertainty is questionable.

The reviewer is correct, that valuable uncertainty estimates are often missing in growth rate studies, and hence we adjusted the statement to "(...) hence the Kalman smoothing will provide valuable insights on the uncertainty related to growth rate measurements"

Line 263 - "modeled the DMA-train instrument numerically" – what does this mean? Do the authors mean that they applied the calibrated instrument kernels to the synthetic data?

We agree with the reviewer that this could be better formulated and hence adjusted the sentence in the following way: "First, we simulated a data set representing a typical nucleation experiment performed in an atmospheric simulation chamber like CLOUD and modelled the DMA-train response according to the above Kernel functions and then applied the FIKS to this synthetic dataset."

Line 272 - simulated and reconstructed size distributions are compared qualitatively as "very good". Give that the y and colour axes are both log scale it is hard for the reader to get a sense of goodness of fit. A more quantitative assessment of the similarity between simulated and reconstructed size distributions is warranted. Graphical representations to more easily demonstrate this would help e.g. size distributions at discrete points in time with uncertainties shown, total number concentrations above given size cuts with uncertainties.

We agree with the reviewer that a more quantitative comparison could be helpful for the manuscript. As detailed comparisons of size distributions at a single instant in time are already shown in Fig. 2, we now included a comparison of the estimated total number concentration $N_{tot}$ for both the original design of the DMA-train and the proposed adjusted kernels in Fig. 6d. This not only demonstrates that the original design already achieves good agreement (we slightly adjusted the wording in the abstract too), but also shows that the new proposed design can achieve even better agreement with less oscillations. This is especially significant in the retrieved growth rates, which was added as Figure 6c for one instant in time. We added/adjusted the following paragraph: "The resulting size-distribution given by FIKS is shown in Fig. 6b. Comparison between the two reconstructions in Fig. 2b and Fig. 6b reveals that the new choice of DMA-channels stabilizes the size-distribution estimate, which is supported by the comparison of the estimated total number concentrations (Fig. 6d). While with the original design the size distribution evolves in step-wise manner when the growing particle mode reaches larger size-channels, the size distribution retrieved with the adjusted kernels is smoother, also temporally, which is especially visible in the evolution of the total number concentration. The improvement is also

significant for the retrieval of the growth rate, where the estimate no longer overshoots as illustrated by the snapshot size dependence of the growth rate for a single instant in time, shown in Fig. 6c."

Regarding the oscillations on size distributions and process rates resulting from the DMA-train channels – is it not possible to apply some smoothing or correction for this?

Yes, smoothing could be done by for example requiring a regularized measurement operator (Voutilainen and Kaipio, 2001) but then the shown method would no longer be identical to Ozon et al. (2020). Any a posteriori smoothing would end up in a biased estimation, and the true variation, e.g. when the nucleation starts, would not be sharp anymore. We adjusted the text at this point slightly "The problem could be approached by application of a regularization scheme in the measurement model (Voutilainen and Kaipio, 2001) or by adjusting the kernels improving the overlap, which will be discussed in more detail in Section 4.5.", citing the corresponding literature and also mention smoothing again in Section 4.5: "It should be noted, that also regularization schemes in the measurement model as proposed by Voutilainen and Kaipio (2001) could provide smoother estimates, but this would need a significant adjustment of the algorithm provided by Ozon et al. (2020) and is hence not implemented here.".

Line 311 - temporal oscillations in calculated growth rate – some of these appear in the PSM growth rate too, but the authors earlier explain these as a product of the discrete size channels of the DMAtrain. So then why are they mirrored in the PSM derived growth rates? Authors note this as "remarkable" and an indication that these are real oscillations on the growth rate. This does not agree with what they mentioned earlier about it being an instrumental artifact in the simulated data. Some work is needed to explain this – what is real and what is an instrumental artifact. And if some of it is real, can the authors suggest why this would occur?

We agree with the reviewer that it is not easy to judge whether oscillations are instrumental or occurring in the chamber. We just base our reasoning on the fact that the oscillations are now not occurring exactly when the size-distribution reaches a new size channel in the DMA-train and second that these oscillations occur in independent instruments sampling from a different port in the CLOUD chamber. We now added the first reasoning to the revised version to clarify this: "In contrast to the oscillations found in the growth rate for the simulated case, the fluctuations do not only occur when the size-distribution reaches a new DMA-train size channel."

Line 331 – could the Henritzi et al size distribution be reproduced here for direct comparison? It would make the paper much more readable and enable direct comparison of results. A direct comparison including errors would again be of more use than two log-scale colour plots.

We agree with the reviewer that it is easier to show the comparison of the size-distribution inverted using the standard approach (following Stolzenburg and McMurry, 2008) with the results from the FIKS, instead of referring to Heinritzi et al. (2020), and Stolzenburg et al. (2020). We therefore also included the comparison of $N_{tot}$ for the standard inversion procedure and the FIKS in new panels for Figures 3-5, demonstrating the solid agreement between the two approaches across the different experiments. We adjusted the text accordingly. The small offset in reconstructed $N_{tot}$ for the organics experiments points towards the fact that there are indeed methodological differences (not-valid Poisson assumption in the

FIKS), however the difference is within a factor of 1.3 much smaller than what is observed for the nucleation rate difference to the PSM estimate, indicating that the different calibration procedures might be the more important offset. We therefore added: "As the deviation in the reconstructed total number concentration of the DMA-train data using two inversion procedures is only a factor of ~1.3 (Fig. 4d), the formation rate discrepancies could be largely due to the more difficult and uncertain calibration procedures.".

Line 336 – While the result presented in fig 4b is indeed an improvement upon inter-instrument differenced of up to an order of magnitude reference from the literature, it is still clear that the nucleation rates derived from the different instrument do not, within the calculated uncertainties, agree. If the reasons for this are well understood from the referenced literature, the author needs to summarize the argument here, and not rely on the reader having detailed knowledge of these other studies from the CLOUD chamber. As it stands the results shown in fig 4b do not support on the of the major claims of this paper – that the FIKS method is in agreement with other methods for calculating process rates. Is it the case that the known uncertainties for the two methods are actually missing a large source of uncertainty? This needs to be addressed. I would like to point out that, even if within known sources of uncertainty these results are not in agreement, as fig 4b suggests, the reduction in disagreement from previous studies is still a very important result that deserves publication and can assist the community to improve our process rate calculations. My point is that the authors must be more rigorous and accurate in describing what has and has not been achieved here.

As also requested by Reviewer #2, we added the DMA-train nucleation rate estimates using the traditional method to the experimental Figures and also discussed the observed discrepancies in the organics experiment in more detail. Please see the discussion there.

Line 340 – This decreasing FIKS growth rate due to a lack of information above 4.3 nm seems problematic. As I understand from the text, there is no information to suggest a decreasing growth rate here, just an assumption of smoothness. Should the algorithm not then be adjusted to take into account where there is not enough information content to calculate a growth rate, and simply not report one, in a similar manner to INSIDE?

The reviewer is correct, this might give an incorrect implication of the size dependence of GR. Thus we modify the GR figures by shading the area where there is no data.

Lines 403-407 - As in previous sections, it is difficult to compare size distributions using a log-scale colour plot. More meaningful comparison, which requires uncertainties, could be achieved using a) snapshots in time, b) cumulative concentrations above given limits. The colour plots shown give a helpful graphical indication of how the different inversions work, which is valuable, but more quantitative evaluation is needed in addition to enable compare between the proposed DMAtrain configurations being discussed here.

We added the total concentration (panel d) and a snapshot of the growth rate at 1h20min (panel c) in order to better compare the improvement due to the adjusted kernels.

Lines 408-418 - This is a really valuable discussion linking instrument design to intended use and I am pleased to see it being brought to the attention of the community through this well explored example.

Thank you! It seemed like a bottleneck for the method that was worth pointing out.

---

## Author Comment (AC2)

**Reviewer replies: Reviewer #2**

The manuscript by Ozon et al. describes the application of a new theoretical method (FIKS, Fixed Interval Kalman Smoother) for deriving new particle formation and growth rates from size distribution measurements. The manuscript that introduces the theoretical framework of the applied method is currently under review [2, Ozon et al. (2020)]. In the present paper the FIKS method is applied to artificially generated size distribution data and data from the CLOUD experiment for three different chemical systems (sulfuric acid + ammonia, highly-oxygenated organic compounds and iodic acid). The data used for FIKS are taken from the DMA-train, which provides time-resolved particle concentrations for seven different size channels (between 1.8 nm and 8 nm, see [4, Stolzenburg et al.,2017]). From these data the FIKS method yields the new particle formation rate (J at 1.7 nm) as a function of time and the particle growth rate (GR) as a function of the particle size. These results are compared with the J and GR derived from other (established) methods. For the derivation of the new particle formation rate the particle number concentrations measured with the Particle Size Magnifier and the Scanning Mobility Particle Sizer are used [1, Dada et al., 2020], whereas for the derivation of the growth rate the INSIDE method with the DMA-train data are used [3, Pichelstorfer et al., 2018]. Overall, the inter-comparison between the results from FIKS and the other methods show good agreement. The FIKS method has the benefit of providing an uncertainty range for the derived quantities. Ozon et al., further demonstrate that the method can be used to optimize the settings of the DMA-train in terms of the set size resolution such that size distribution can be reconstructed by the proposed method with high accuracy. This is very important as such a guideline can improve the data quality in further experiments.

Overall, I agree with the authors that the development of sophisticated data evaluation methods as the one presented here, is important and lacks somewhat behind the instrument development. Therefore, I highly favor the publication of the present study and have only a few suggestions for further improvement and clari_cation (listed below). What I find, however, somewhat problematic is the fact that the method paper [2, Ozon et al. (2020)] is not finally published yet. In this respect, I would also like to mention that I did not review the method itself but only its application in the present study. Therefore, I think that it would be appropriate to wait for the final publication of [2, Ozon et al. (2020)] before the present manuscript can be published in ACP.

We thank the reviewer for his/her thoughtful replies and are happy that he/she is convinced about the importance of this manuscript. We are happy to inform that the method paper has been published. Please see link: https://gmd.copernicus.org/articles/14/3715/2021/gmd-14-3715-2021.html.

Page 1, line 33/34: The authors mention here that \potentially crude approximations" can be made when formation and growth rates are derived. It would be good to explain what approximations are meant here.

We removed the statement about potentially crude approximation, because "rather simple regression or balance equation approaches" is exactly what we mean.

Page 2, line 72/73: I think, the data from the DMA-train can also be used to derive new particle formation rates (from the smallest size channel) and that these data need to be included in the further analysis and discussion. This could show whether the differences in J from the PSM and the FIKS method arise mainly due to the use of data from two different instruments or from the different methods.

We have added the nucleation rates estimated with the DMA-train data and the traditional method for all three cases. In the sulfuric acid case and the iodic acid case, the nucleation rate values are quite similar to the two other approaches, supporting the claim that we should be quite confident about the estimations. The observation that in all test cases the direct DMA-train-based estimates and their credible intervals are somewhat in the middle of those given by FIKS and the PSM, hints that the differences between the FIKS and PSM estimates are caused by the use of different instruments and the differences in the estimation methods.

Page 7, line 218 : Here it is mentioned that the number of the size channels is 32. Some discussion should be included how this choice affects the outcome of the results. Can a larger number of size channels improve the agreement between the FIKS method and the other methods?

We agree that this needs a short discussion and added the following statement to the manuscript: "We use a resolution of 32 bins from 1.7 to 10 nm for the FIKS to keep the computational effort low. We tested also 16 to 64 size discretization bins, but higher resolution required additional adjustments in the size-correlation of the covariance given in Eq. (8), which would result in significant differences compared to the original work of Ozon et al. (2020) without providing significantly more accuracy."

Page 11, line 335 : The discrepancy between J from the PSM and the FIKS method is a factor of 2.5, i.e., it is significantly larger as for the sulfuric acid/ammonia and the iodic acid system. I would like to see some further discussion on the possible reasons for this difference.

We agree with the reviewer that the organics experiment shows the largest discrepancies and this should be further clarified. There is a plausible reason for the deviations in the organic experiment compared to the other two: In the organic case, there are the lowest number concentrations in the chamber for all three experiments. These are already close to the limit of detection in the DMA-train and hence our uncertainties are not well described by the Gaussian assumptions inherent to the FIKS, but Poisson statistics are needed. Further, we know that our calibrations for organics both for the PSM and for the DMA-train are quite uncertain, and derived with different approaches. For the PSM we use the method of Dada et al. (2020) by comparing with the NAIS, while for the DMA-train we used a lab-based calibration with beta-caryophyllene particles (Wlasits et al., 2020) which are not exactly equal to the alpha-pinene particles. Hence, it is not surprising that there are some deviations caused by the instruments and not the methods, which can be seen from Figure 4b) where we also included the traditional nucleation rate approach using DMA-train data (see earlier comment). We thus added: "Also the formation rate retrieved with the method from Dada et al. (2020) but using the DMA-train data is significantly higher, but in-between the two estimates. The possible deviation has hence two plausible reasons: The instrumental differences can be caused by different calibration procedures for the DMA-train and PSM (Dada et al. (2020) direct cross-calibration using NAIS versus Wlasits et al., (2020) using beta-caryophyllene ozonolysis surrogates). And the methodological differences could arise from the very low counting statistics in the DMA-train during this experiment compared to the other two, which will cause the inherent Gaussian assumption of the FIKS to fail."